# Cytotoxic T cells are able to efficiently eliminate cancer cells by additive cytotoxicity

Bettina Weigelin [1,2,3,4 ✉], Annemieke Th. den Boer[5], Esther Wagena[1], Kelly Broen[6], Harry Dolstra[6], Rob J. de Boer [7], Carl G. Figdor [8], Johannes Textor[8] & Peter Friedl [1,2,9 ✉]

Lethal hit delivery by cytotoxic T lymphocytes (CTL) towards B lymphoma cells occurs as a binary, "yes/no" process. In non-hematologic solid tumors, however, CTL often fail to kill target cells during 1:1 conjugation. Here we describe a mechanism of "additive cytotoxicity" by which time-dependent integration of sublethal damage events, delivered by multiple CTL transiting between individual tumor cells, mediates effective elimination. Reversible sublethal damage includes perforin-dependent membrane pore formation, nuclear envelope rupture and DNA damage. Statistical modeling reveals that 3 serial hits delivered with decay intervals below 50 min discriminate between tumor cell death or survival after recovery. In live melanoma lesions in vivo, sublethal multi-hit delivery is most effective in interstitial tissue where high CTL densities and swarming support frequent serial CTL-tumor cell encounters. This identifies CTL-mediated cytotoxicity by multi-hit delivery as an incremental and tunable process, whereby accelerating damage magnitude and frequency may improve immune efficacy.

[1] Department of Cell Biology, RIMLS, Radboud University Medical Center, Nijmegen, The Netherlands. [2] David H. Koch Center for Applied Research of Genitourinary Cancers, Department of Genitourinary Medical Oncology, The University of Texas MD Anderson Cancer Center, Houston, TX, USA. [3] Department of Preclinical Imaging and Radiopharmacy, Eberhard Karls University, Tübingen, Germany. [4] Cluster of Excellence iFIT (EXC 2180) "Image-Guided and Functionally Instructed Tumor Therapies", University of Tuebingen, Tübingen, Germany. [5] Department of Internal Medicine, Maastricht University, Maastricht, The Netherlands. [6] Department of Laboratory Medicine – Laboratory of Hematology, Radboud University Medical Center, Nijmegen, The Netherlands. [7] Theoretical Biology and Bioinformatics, Utrecht University, Utrecht, The Netherlands. [8] Department of Tumor Immunology, RIMLS, Radboud University Medical Centre, Nijmegen, The Netherlands. [9] Cancer Genomics Centre Netherlands (CGC.nl), Utrecht, The Netherlands. ✉email: bettina.weigelin@med.uni-tuebingen.de; peter.friedl@radboudumc.nl

Cytotoxic T lymphocytes (CTLs) execute effector function by a cyclic process of transient cell–cell interaction and paracrine delivery of cytotoxic effector molecules to target cells, followed by target cell death[1,2]. CTL and natural killer (NK) cells can bind to and attack more than one target cell sequentially[3–5]. As a consequence of multi-hit delivery, single CTLs are capable of eliminating multiple antigenic target cells in vitro and in vivo, ranging from 1 up to 20 kills per CTL and day, as estimated from bulk killing assays and mathematical modeling[6,7]. Such high efficacy of CTL-mediated serial killing, however, is rarely observed in patients receiving adoptive transfer of tumor-specific T cell receptor (TCR)-engineered or chimeric antigen receptor (CAR) T cells[8]. Evidence for effective serial target cell killing after 1 : 1 pairings has further not been obtained in antigenic solid tumor models in mice where CTLs were found to form predominantly short-lived interactions with target cells, which rarely result in direct apoptosis induction[9–13]. Beyond tumor models, non-productive CTL interactions and failed target cell eradication are also observed in alloimmune response against transplants[14] and murine cytomegalovirus-infected cells in the mouse dermis[15]. Consequently, the physiological relevance of CTL contacts, which—individually—fail to induce target cell death remains unclear.

Preclinical and emerging clinical data indicate that successful CTL effector function correlates with high local CTL densities[16,17]. CTL density control of killing efficacy has further been proposed by mathematical modeling based on bulk three-dimensional (3D) killing assays[18]. Different mechanisms have been proposed how high CTL densities may mediate effective killing. High CTL densities enable high contact frequencies and multiple CTL–tumor cell encounters. This may increase the likelihood of multiple CTL simultaneously attacking a single target cell[19,20] and the probability for tumor cells to be contacted by rare potent CTL[18]. In addition, cytotoxic cytokine release, including interferon-γ (IFNγ) and tumor necrosis factor, is positively associated with high-density T cell infiltration[21]. However, whether individually ineffective CTL interactions in solid tumors can become effective at high CTL density and by which mechanism individually "ineffective" contacts contribute to target cell killing remains to be investigated.

Here we apply live-cell microscopy in vitro and in vivo, introduce molecular damage reporters into antigenic tumor cells, and analyze the types, temporal mechanisms, and thresholds of damage induced in antigenic tumor cells by "seemingly ineffective" CTL contacts. These data indicate that individually ineffective CTL contacts induce sublethal damage, which becomes integrated over time in the target cell until apoptosis is induced. Through a multi-hit mechanism, CTL induce tumor cell death when density is high, whereas tumor cells repair sublethal damage and survive when CTL density is low. The mechanism of "additive cytotoxicity" explains how individually ineffective interactions become effective at high CTL density and can be utilized to improve therapy success by microenvironmental or immune modulation.

## Results

**CTL serial conjugation and effector function.** In vitro-activated chicken ovalbumin (OVA)-specific OT1 CTLs were confronted with transformed mouse embryonic fibroblasts expressing the OVA peptide (MEC-1/OVA) and the co-stimulatory molecule B7.1[22,23]. In contrast to tumor cells that evolved in vivo, this engineered model lacks natural immune escape modifications (e.g., downregulation of major histocompatibility complex-I or apoptosis resistance) and, thus, represents an idealized model for maximized CTL efficacy at a single cell level. After 30 h of co-culture, killing efficacy was near 100% at high effector–target (ET)

ratios and reached background level below ET ratios of 1 : 128 (Supplementary Fig. 1a). To assess the serial killing efficacy of individual CTL, we analyzed antigen-specific CTL–target cell interactions and outcome in long-term time-lapse microscopy recordings by tracing individual CTL during interaction with target cells. The duration of individual CTL–target cell interactions was variable (lasting minutes to hours), with lag times between initial CTL binding and target cell death lasting 1.8 ± 1.5 h, and was followed by a subsequent period of ongoing CTL engagement with the dead cell body ("necrophilic phase") (Supplementary Fig. 1b–e). This extended lag phase until apoptosis differs from lag phases obtained for CTL-mediated killing of target cells from leukocyte lineages, lasting <5–25 min[24–26]. In regions of low local CTL density, sequential contacts by an individual CTL resulted in the serial killing of multiple neighboring target cells (Fig. 1a). On the population level, 50% of the CTL acted as serial killers (maximum of 11 killed target cells/24 h), whereas a small CTL subset (15%) repeatedly contacted target cells without inducing apoptosis (Supplementary Fig. 1f). The percentage of CTL with killing capacity correlated with the surface expression of LAMP-1 by 85–90% of CTL, indicating recognition of the target cells and lytic vesicle exocytosis by the majority of CTL (Fig. 1b and Supplementary Fig. 1g). The lag phase to apoptosis was neither compromised nor accelerated over consecutive killing events (Fig. 1c), which resulted in a consistent eradication frequency of 1 kill every 2 h (Supplementary Fig. 1h). This excludes gain of cytotoxicity by kinetic priming through repetitive antigenic interactions. Thus, OT1 CTL serially eliminate highly immunogenic target cells over 24 h and in a non-exhaustive manner.

**CTL induce sublethal plasma membrane damage.** To compare effector function against solid tumor cells, which typically retain resistance to CTL-mediated killing, OT1 CTL were confronted with mouse melanoma B16F10 cells expressing the OVA peptide (B16F10/OVA) (Supplementary Fig. 2a–d and Supplementary Movie 1). As a second model, interleukin-2 (IL-2)-activated human SMCY.A2 CTL[27], which recognize an HLA-A2-restricted antigen encoded on the y chromosome, were confronted with male human melanoma cell lines BLM or MV3 (Supplementary Fig. 2e–i and Supplementary Movie 1). Compared to the MEC-1/OVA cells, these three melanoma models show delayed, but ultimately effective target cell elimination at the assay endpoint after 24 h, whereas OVA-negative B16F10 or female MCF-7 cells survived (Fig. 1d). Thus, the endpoints of both mouse and human models for probing CTL effector function show comparable target cell elimination in 3D culture. Notably, across all cell types tested, only a minority of individual CTL–target cell contacts induced apoptosis at first encounter, whereas 60–70% (MEC-1/OVA, BLM) or >90% (B16F10/OVA, MV3) of individual conjugations were followed by target cell survival (Fig. 1e).

CTL degranulation induces transient perforin-mediated pores in the target cell membrane, which facilitates diffusion of extracellular factors into the target cell, including CTL-derived granzyme B[28]. OT1 CTL deficient in perforin expression failed to kill B16F10/OVA cells in organotypic culture (Supplementary Fig. 2j), whereas CTL effector function remained intact after interference with Fas–FasL interaction (Supplementary Fig. 2k). Further, adding perforin-deficient OT1 CTL to a fixed number of wt OT1 CTL did not increase killing efficiency, indicating that perforin-independent mechanisms delivered by excess CTL, including soluble mediators, do not induce or enhance cytotoxicity against B16F10/OVA cells (Supplementary Fig. 2l). Thus, elimination of B16F10/OVA cells by OT1 CTL in 3D culture critically depends on perforin.

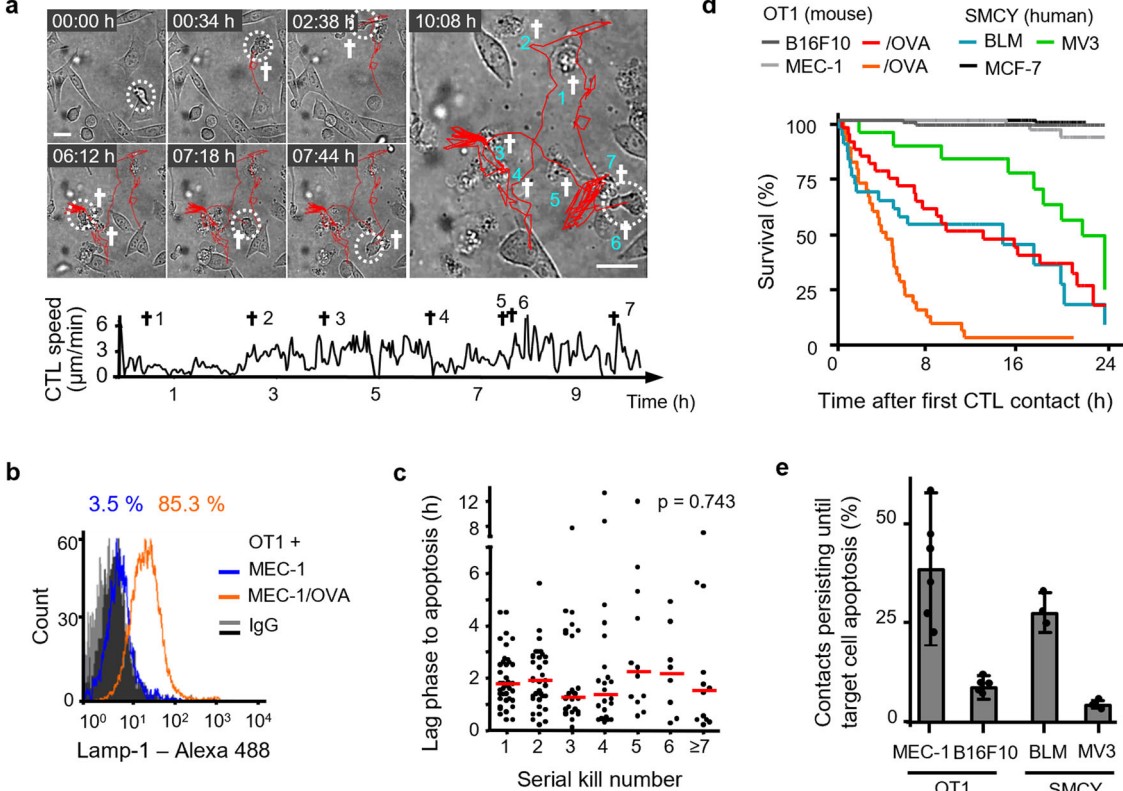

**Fig. 1 CTL serial conjugation and effector function in different tumor models. a** Time-lapse sequence and migration track of one OT1 CTL killing seven MEC-1/OVA target cells sequentially within 11 h. Circles, CTL; cross, apoptotic target cell; scale bar, 20 µm. Representative example of a serial killer CTL derived from analysis of 43 CTL pooled from 8 independent experiments). **b** Lamp-1 expression at the surface of OT1 CTL after 24 h of 3D co-culture with MEC-1/OVA cells. Representative example from three independent experiments. **c** Lag phase until apoptosis of consecutive single interaction kills by the same CTL (43 CTL from 8 independent experiments). Red bars, median. *p*-Value, Kruskal–Wallis test. **d** Population survival of four antigen-dependent mouse and human target, and three control cell lines. Quantifications from three independent experiments for each cell line. B16F10, 98 cells; B16F10/OVA, 31 cells; MEC-1, 57 cells; MEC-1/OVA, 32 cells; BLM, 28 cells; MV3, 18 cells; MCF-7, 102 cells. **e** Inefficiency of inducing apoptosis by individual CTL contacts in OT1 and SMCY.A2 CTL models. Error bars, mean ± SD obtained from 104 MEC-1/OVA, 183 B16F10/OVA, 53 BLM, and 50 MV3 contacts from $N = 5$ (MEC-1/OVA, B16F10/OVA) or 3 (BLM, MV3) independent experiments per cell model. Source data are provided as a Source Data file.

To visualize perforin pore formation and to discriminate sublethal cytotoxic hits from functionally inert interactions, MEC-1/OVA and B16F10/OVA cells were engineered to express the calcium sensor GCaMP6s[29], and were monitored for transient $Ca^{2+}$ influx, as a proxy for perforin pore formation and recovery[30] (Fig. 2a1, Supplementary Fig. 3a, and Supplementary Movie 2). $Ca^{2+}$ influx induction was strictly dependent on perforin expression in OT1 cells (Fig. 2b; "perforin events") and the $Ca^{2+}$ signal originated at CTL–tumor cell contact region differed in signal intensity and duration from unspecific intracellular $Ca^{2+}$ fluctuations (Supplementary Fig. 3b1 and b2, c). A substantial fraction (40%) of antigen-specific but nonlethal CTL–target cell contacts showed CTL-associated perforin events in B16F10/OVA cells (Fig. 2b), which were transient (median: 30 s; Fig. 2c) and in 90% followed by target cell survival (Fig. 2d).

To test whether the variability of perforin events in B16F10/OVA cells were a consequence of heterogeneous TCR engagement, we quantified $Ca^{2+}$ signaling in CTL upon target cell contact. OT1 CTL showed comparably high rates of $Ca^{2+}$ signaling when contacting MEC-1/OVA and B16F10/OVA cells (80–85%), typically within seconds after contact initiation (Fig. 3a). When co-registered with perforin events in target cells, 40% of $Ca^{2+}$ positive CTL contacts with B16F10/OVA coincided with or were immediately followed by a perforin event in the target cell (Fig. 3b, c). In conclusion, although TCR triggering in OT1 CTL occurs reliably, the induction of perforin events in the target cell varied.

**CTL-induced structural damage in target cells**. To address whether transient perforin pores were associated with structural intracellular damage, B16F10/OVA cells were engineered to express green fluorescent protein tagged with a nuclear localization signal (NLS-GFP)[31] or 53BP1trunc-Apple[32]. NLS-GFP leakage into the cytoplasm was detected in 25% of CTL–target cell contacts and was absent when CTL lacked perforin expression (Fig. 2a2, b, Supplementary Fig. 3d–g, and Supplementary Movie 3). Recovery, as indicated by diminishing NLS-GFP signal in the cytosol and recovery in the nucleus, occurred in 75% of events within minutes to hours (median: 49 min; Fig. 2c, d). 53BP1 initiates DNA damage repair complexes, which can be visualized as repair foci by 53BP1trunc-Apple[32] (Supplementary Fig. 3h and Supplementary Movie 4). 53BP1 foci were induced in 35% of CTL contacts, in dependence of perforin expression in OT1 CTL (Fig. 2a3, b) and CTL density (Supplementary Fig. 3i, j). CTL-induced 53BP1 foci persisted for several hours (median: 4 h) and were resolved in 73% of events (Fig. 2c, d). These data indicate that CTL contacts induce reversible sublethal damage to the nuclear lamina and DNA.

**Death induction by multiple CTL**. Across all tested tumor models, sequential or simultaneous contacts by multiple CTL with the same target cell occurred before target cell death (Fig. 4a). In the B16F10 model, >90% of successful kills were preceded by two to nine CTL encounters by distinct CTLs

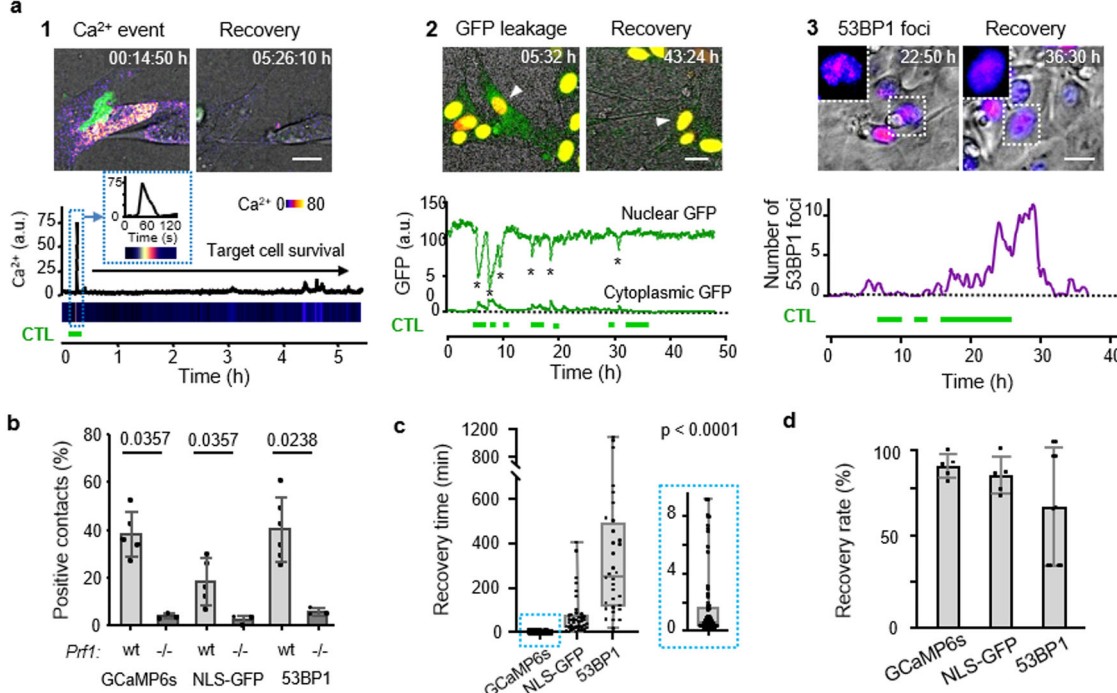

**Fig. 2 Type and kinetics of sublethal damage induced by CTL. a** Reporter strategies. Green horizontal bars, duration of CTL contacts. Panel 1: CTL-mediated perforin pores visualized as $Ca^{2+}$ influx into B16F10/OVA target cells using the GCaMP6s reporter. Time-resolved intensity plot of GCaMP6s event in the target cell cytosol followed by survival. Green, OT1 CTL (GFP); Fire LUT, $Ca^{2+}$ level (GCaMP6s). Panel 2: CTL-mediated structural damage of the nuclear lamina monitored as NLS-GFP leakage into the cytosol. Nuclear and cytosolic GFP intensity plotted over time. Green, NLS-GFP; red: Histone-2B-mCherry; arrowhead: monitored cell. Asterisks, nuclear leakage events. Panel 3: CTL-mediated DNA damage response plotted as 53BP1trunc-Apple focalization in the nucleus over time. Fire LUT, 53BP1trunc-Apple. Insets, zooms of reversibility of 53BP1 foci. Scale bars, 10 µm. Image sequences in panels 1 and 2 show representative examples from experimental datasets shown in **b–d**. **b** Percentage of contacts of wt and perforin-deficient CTL with B16F10/OVA cells inducing $Ca^{2+}$ events, NLS-GFP cytosolic leakage, and 53BP1trunc-Apple foci in target cells. Data show the mean ± SD from $N = 3$ (GCaMP6s/wt, NLS-GFP/wt/prf1−/−), 5 (53BP1trunc-Apple/wt), and triplicate movies from 1 (GCaMP6s/prf1−/−, NLS-GFP/prf1−/−) independent experiments. p-Values, two-tailed Mann–Whitney test comparing wt and prf1−/− datasets. **c** Recovery times from initiation to termination of GCaMP6s, NLS-GFP, and 53BP1trunc-Apple reporter activity. Data show the medians with whiskers from minimum to maximum values derived from $N = 32$ (53BP1trunc-Apple), 40 (NLS-GFP), and 71 (GCaMP6s) individual events pooled from 3 (NLS-GFP, GCaMP6s) and 5 (53BP1trunc-Apple) independent experiments. p-Value, Kruskal–Wallis test comparing all groups corrected by Dunn's multiple comparisons test. **d** Percentage of cells with sublethal damage event, which fully resolved. Data show the mean ± SD (five independent experiments per reporter) for an 1:2 ET ratio. Source data are provided as a Source Data file.

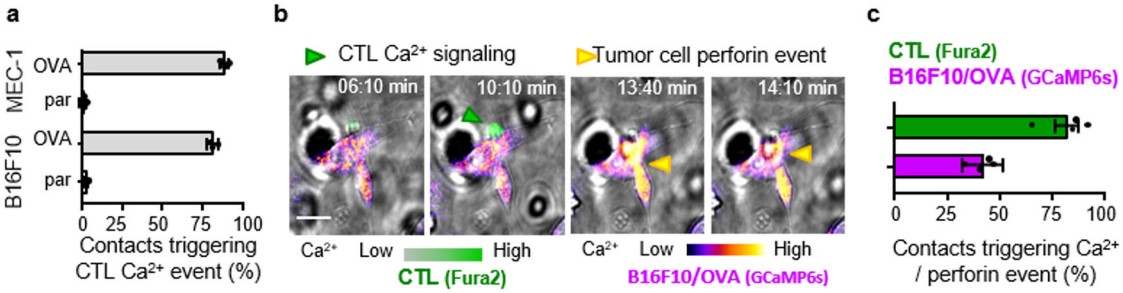

**Fig. 3 Correlation between CTL Ca2+ signaling and perforin pore formation in different target cells. a** Percentage of CTL contacts associated with $Ca^{2+}$ events in OVA-expressing target cells (OVA) compared to OVA-negative parental (par) cells. Data show the means ± SD from three independent experiments. **b** Image sequence of Fura2-labeled OT1 CTL during contact with GCaMP6s-expressing B16F10/OVA cell. Arrowheads: green, Fura2-positive event; yellow, GCaMP6s-positive event. Scale bar, 10 µm. **c** Percentage of contacts triggering $Ca^{2+}$ events in OT1 CTL or B16F10/OVA target cells, respectively. Quantification (means and SD) was performed by manual analysis from 78 contacts pooled from $N = 4$ independent experiments. Source data are provided as a Source Data file.

(Fig. 4b). When co-registered over time for each target cell, death induction was preceded by serial perforin events with variable onset and frequency between hits. The lag time between contact initiation and first perforin event was 6 min in MEC-1/OVA cells and 16 min in B16F10/OVA cells (Fig. 4c). Killing of the MEC-1/ OVA cells was preceded by perforin events with 4 min median interval, whereas perforin events in B16F10/OVA cells occurred with longer intervals of 18 min (Fig. 4d). Although >50% of contacts induced only one perforin event in B16F10/OVA cells, in 40% of CTL contacts yielded at least two sequential perforin

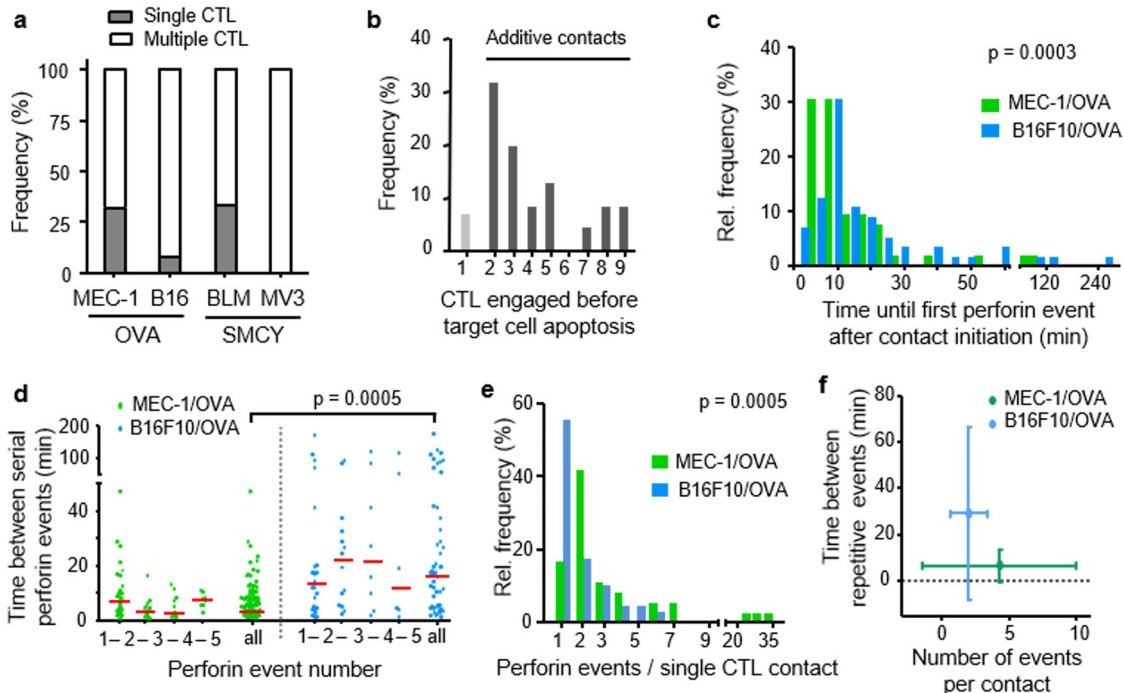

**Fig. 4 Kinetics and frequency of CTL-induced sublethal events. a** Percentage of apoptosis events preceded by multiple or single CTL contacts in mouse and human melanoma models. Pooled data representing ≥100 contacts from ≥3 independent experiments per cell line. **b** Percentage of CTL engaged before target cell death in B16F10/OVA co-culture with OT1 CTL. **c** Lag phase until first $Ca^{2+}$ event after contact initiation in MEC-1/OVA and B16F10/OVA target cells. Data from 55 (B16F10) and 52 (MEC-1) $Ca^{2+}$ events. *p*-Value, two-tailed Mann–Whitney test. **d** Intervals between sequential $Ca^{2+}$ events in the target cell induced by the same CTL in one single contact. Medians were 4 min (MEC-1/OVA) and 18 min (B16F10/OVA). Red line, median. Data show 53 (B16F10) and 119 (MEC-1) $Ca^{2+}$ events. *p*-Value, two-tailed Mann–Whitney test. **e** Number of $Ca^{2+}$ events associated with the same CTL. Data show 55 (B16F10) and 36 (MEC-1) CTL. Data from **b** to **d** pooled from three (B16F10) and two (MEC-1) independent experiments. *p*-Values, two-tailed Mann–Whitney test. **f** Number of $Ca^{2+}$ events per CTL contact plotted against frequency of sequential $Ca^{2+}$ events in MEC-1/OVA cells compared to the B16F10/OVA cells. Data from 53 (B16F10) and 119 (MEC-1) $Ca^{2+}$ events pooled from 3 (B16F10/OVA) and 2 (MEC-1/OVA) independent experiments. Source data are provided as a Source Data file.

events (Fig. 4e). Thus, compared to MEC-1/OVA cells, delayed killing of B16F10/OVA cells correlated with delayed delivery of perforin events (Fig. 4f).

**Additive cytotoxicity**. We then addressed whether apoptosis could have been induced by rare deadly CTL hits ("stochastic killing") or, instead, by sublethal contacts that add up over time ("additive cytotoxicity") (Fig. 5a and Supplementary Movies 2 and 5). Therefore, we analyzed whether the lethality of the final hit was enhanced by, or independent of, previous perforin events and plotted the lag time to apoptosis together with target cell survival probability relative to the number of pre-final perforin events. Target cells which received two or more hits prior to the lethal one showed accelerated apoptosis induction, together with a sharp decrease in survival probability (Fig. 5b and Supplementary Fig. 4a, b). The dependence of the lag time to apoptosis on prior CTL hits is expected when additive effects exists, but it is inconsistent with the stochastic killing hypothesis. To explore stochastic killing directly, we performed the same analysis on simulated data, using randomly swapped times between hits and between the last hit and apoptosis. Here, cell death induction was gradual and neither the lag time to apoptosis nor the survival probability was dependent on the number of prior hits (Fig. 5c).

To address how long sublethal events remain relevant, we estimated the time required to repair the damage caused by a single perforin event, using a Cox regression model based on additive killing. This resulted in an estimated damage half-life of 56.7 min (95% confidence interval: 33.1–112.2) of single perforin events to contribute to lethal outcome (Fig. 5d). This interval was

consistent with the recovery kinetics of nuclear lamina defects (Fig. 2c). Using the Bayesian Information Criterion (BIC) to compare model fits showed that the model which integrates serial damage and decay explained the data better (BIC difference > 10) than a model based on the number of perforin events alone.

To integrate timing and heterogeneity of damage, we extended our model to allow for variable damage delivered by serially engaging CTL. A multivariate Cox model shows that perforin events associated with the last interacting CTL (hazard ratio: 3.7, Wald's test $p < 10^{-7}$) and earlier CTL in contact (hazard ratio: 3.0, $p = 0.001$) are both important predictors of apoptosis. This analysis shows no statistical benefit of distinguishing between last and prior CTL, indicating that their integrated, additive effects are relevant for target cell death. A simpler Cox model that considers all CTL equally relevant is a more parsimonious description (as indicated by the lowest BIC) of the data than models that focus on the last interacting CTL alone (BIC difference: 6.6). In addition, a model that assumes variable damage between prior and last CTL contacts was inferior in explaining the data (BIC difference: 2.8). Thus, from a statistical perspective, our current data offers no support for the hypothesis that a rare lethal hit leads to target cell elimination.

Further, the majority (60%) of single CTL interactions that preceded target cell death were associated with multiple perforin events (Supplementary Fig. 4c, d). This highlights that both single and sequential CTL contacts can deliver multiple perforin hits. When co-registered, prolonged duration of interactions correlated with an increased number of delivered perforin events for either lethal or nonlethal outcome (Supplementary Fig. 4e). In

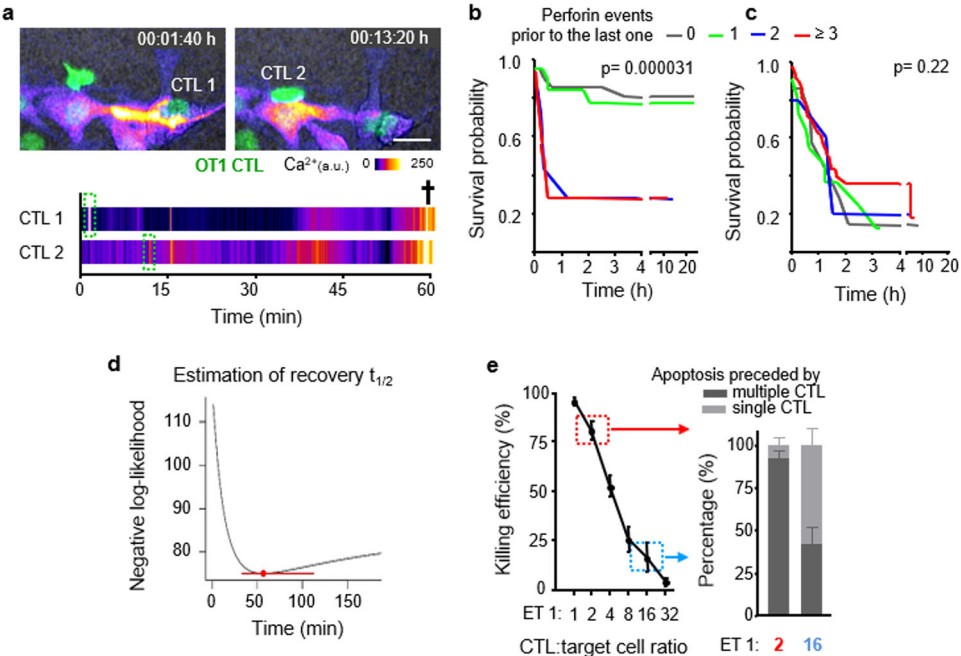

**Fig. 5 Additive cytotoxicity and estimation of damage recovery half-life. a** Time-lapse sequence and intensity plot of multiple $Ca^{2+}$ events followed by target cell apoptosis. Green fluorescence, OT1 CTL (dsRed); Fire LUT, $Ca^{2+}$ intensity (GCaMP6s). Cross, target cell death; Scale bar, 20 μm. Image sequence shows a representative example from 124 perforin events preceding $N = 63$ B16F10/OVA apoptosis events, pooled from 5 independent experiments. **b** Survival probability of B16F10/OVA cells having received increasing numbers of $Ca^{2+}$ events. **c** Simulation of stochastic apoptosis induction by permutation of waiting times between $Ca^{2+}$ events, survival, and lag times until apoptosis. *p*-Values in **c**, **d**, two-sided log rank test comparing all groups. **d** Estimation of damage recovery half-life in B16F10/OVA after one single $Ca^{2+}$ event by a statistical model that assumes additive killing (see "Methods"). Point and error bar: damage half-life that is most consistent with the data and at 95% confidence interval. Data in **b**–**d** represent 124 perforin events related to $N = 63$ B16F10/OVA apoptosis events, pooled from 5 independent experiments. **e** Killing efficacy of B16F10/OVA cells and percentage of preceding single or multiple interacting CTL in dependence of ET ratio. Left panel, means and SD from $N = 2$ independent experiments; right panel, 110 contacts from $N = 4$ independent experiments per ET ratio. Source data are provided as a Source Data file.

aggregate, these results indicate that sequential sublethal events, which are delivered by single or multiple CTL in a timely manner, favor-efficient target cell killing.

**Multi-hit delivery as a function of CTL density.** To test whether apoptosis induction under conditions of high CTL density facilitated additive, multi-hit interactions or, instead, a higher probability of lethal single-hit interactions of few CTL[18], we titrated CTL density and monitored individual CTL–target cell interactions by time-lapse microscopy. At high CTL density (ET 1 : 2) target cell death was frequent and preceded by multiple CTL interactions, whereas at low CTL density (ET 1 : 16) infrequent apoptotic events were predominantly preceded by single-contact engagements (Fig. 5e). The CTL density effect was not enhanced by the addition of perforin-deficient CTL (Supplementary Fig. 2l). This indicates that high CTL density ("swarming") enables efficient apoptosis induction by favoring serial perforin-dependent hits, preferentially by different CTL, whereas the ineffective killing at low CTL density largely relies upon single encounters.

**Sublethal hit delivery in vivo.** We lastly addressed whether multiple encounters by CTL mediate anti-tumor cytotoxicity in vivo. Activated OT1 CTL were adoptively transferred into C57BL/6J mice bearing intradermal B16F10 melanoma and monitored longitudinally by intravital microscopy through a skin window (Supplementary Fig. 5a, b). A single application of OT1 CTL caused a transient, dose-dependent growth delay of the OVA antigen-expressing but not of control tumors (Supplementary Fig. 5c). B16F10 tumors invade the dermis as multi-cellular strands[33].

Concomitantly, OT1 CTL first accumulated in the tumor periphery and subsequently redistributed towards the invasive tumor front (Fig. 6a). This resulted in local ET ratios of 1 : 1 along the tumor–stroma interface, whereas ET ratios in the tumor core remained below 1 : 250 (Supplementary Fig. 5d, e). To identify where and by which contact mechanism eradication of tumor cells was achieved, CTL contacts and outcome were mapped using histone-2B/mCherry (H2B/mCherry) to detect nuclear fragmentation in B16F10/OVA cells[12]. Despite comparable ET ratios, high fragmentation rates occurred in the invasion niche but not the tumor rim (Fig. 6b). In either zone, >95% of CTL contacts were transient, short-lived (median duration 15 min), and nonlethal but accumuulated over multiple contacts to contact durations reaching 1 up to several hours (Fig. 6c). When aggregated, >86% of apoptotic events were preceded by multiple CTL contacts and a minority (<14%) by individual CTL conjugation (Fig. 6d). In the tumor invasion niche, high local CTL density coincided with confined migration along tissue structures with enhanced speed compared to the main tumor mass (Supplementary Fig. 6a, b). This supported frequent contacts to B16F10/OVA cells with cumulative contact duration reaching >1 h and nuclear fragmentation in tumor cells in a time-dependent manner (Fig. 6e and Supplementary Movie 6).

To discriminate functional from non-functional, irrelevant interactions, we analyzed the occurrence of CTL-mediated $Ca^{2+}$ events using GCaMP6s-expressing tumors. Most $Ca^{2+}$ events (80%) in B16F10/OVA cells were associated with CTL contacts, but rare without interacting CTL (Supplementary Fig. 6c, d). In invasive tumor subregions with high apoptosis rates, both the frequency of CTL contacts inducing $Ca^{2+}$ events and the tumor

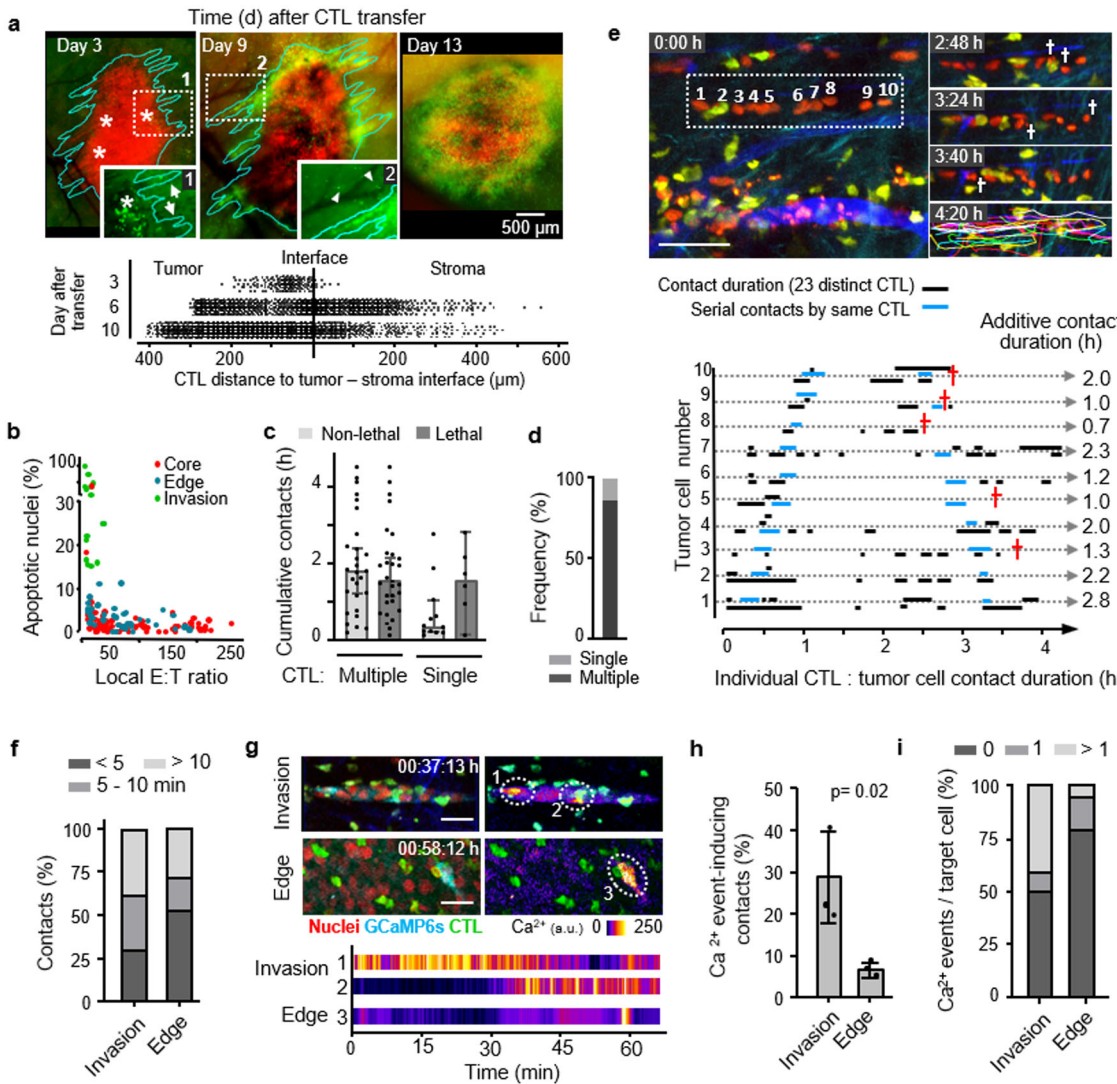

**Fig. 6 Additive cytotoxicity in live tumors detected by intravital multiphoton microscopy. a** Time-dependent CTL accumulation along the tumor–stroma interface. Red, nuclei B16F10/OVA cells; green: OT1 CTL. Lower panel, position of individual CTL. Representative example of one tumor. **b** Correlation of CTL density and apoptotic frequency in tumor subregions. Data show 135 apoptosis events pooled from 8 independent mice. **c** Cumulative contact duration and outcome of single or multiple CTL contacting B16F10/OVA cells. Data represent the median with whiskers from 25 to 75 percentile values. **d** Frequency of apoptosis induction associated with single or multiple interacting CTL. Data in **c**, **d** represent $N = 40$ nonlethal and 37 apoptotic events from 150 h of movies pooled from 20 independent tumors. Error bars, mean ± SD. **e** Representative micrographs and quantification of serial engagements of multiple CTL with B16F10/OVA cells and outcome. Red cross, target cell apoptosis. Ten target cells from one tumor. **f** Percentage of contacts between CTL with B16F10/OVA cells and duration category in the invasion zone vs. tumor edge. **g** Images from time-lapse recordings of GCaMP6s activity in B16F10/OVA target cells in distinct tumor subregions. Dotted circles indicate the example areas plotted for GCaMP6s intensity in the graph below. Image sequence shows a representative example from the dataset analyzed in **h**, **i**. **h** Percentage of CTL contacts that induced one or more $Ca^{2+}$ events in invasion zones vs. CTL-rich subregions at the tumor edge. **i** Percentage of tumor cells receiving none, one, or more than one $Ca^{2+}$ event within a cumulative observation time of 3 h per tumor subregion. Data in **f**, **h**, **i** represent 228 contacts from $N = 4$ independent mice. Data in **h**, mean ± SD (3 positions per subregion from $N = 3$ independent mice). *p*-Value, two-tailed Mann–Whitney test. Scale bars **e**, **g**, 50 μm. Source data are provided as a Source Data file.

cell fraction receiving multiple $Ca^{2+}$ events were by fivefold increased, compared to the non-invading tumor rim (Fig. 6g, h and Supplementary Movies 7 and 8). Thus, in vivo eradication of tumor regions by CTL is a function of local CTL density and frequent sublethal interactions.

## Discussion

When benchmarked against the reliable and near-deterministic elimination of B cells[24–26], CTL effector function against non-hematological solid tumor cells is inefficient. Despite reliable CTL activation by the OVA antigen, only a minority of individual

CTL–tumor cell contacts (~5%) achieves B16F10 melanoma cell killing directly. Instead, 95% of the death events required a sequence of sublethal hits, ideally at least three hits delivered with <50 min recovery intervals in between. These results indicate that CTL-mediated apoptosis induction is not a binary event but may depend on the probabilistic accumulation of damage within the target cell over time. CTL-induced additive cytotoxicity may, hence, follow damage rules of other types of cell death, including the accumulation of (i) DNA double-strand breaks during fractionated low-dose irradiation[34,35], (ii) nuclear damage events in invading tumor cells moving through mechanically challenging confinement[31], or (iii) with much longer time integration,

somatic mutations preceding neurodegeneration[36]. We thus suggest accumulating sublethal damage as the prevailing mechanism of tumor cell eradication by antigen-specific CTL.

In OT1 CTL interactions with B16F10/OVA melanoma target cells, subsequent to perforin-mediated pores in the plasma membrane, we here identified nuclear envelope rupture and DNA double-strand breaks as sublethal damage types. Nuclear envelope proteins including B-type lamins, which form and mechanically stabilize the nuclear lamina, are substrates of Granzyme B[37]. Granzyme B further cleaves inhibitor of caspase-activated DNase (CAD), resulting in the release of CAD, which translocates into the nucleus and induces DNA double-strand breaks[38]. Importantly, rather than leading to cell death directly, CTL-induced damage to both the nuclear lamina and DNA was typically transient, reversible, and followed by tumor cell survival.

The entry of extracellular $Ca^{2+}$ through perforin pores triggers rapid membrane repair mediated by membrane delivery and fusion of cytosolic vesicles at the damage site[39,40]. The recovery times following intracellular structural damage, observed using live-cell reporter systems in target cells, are in agreement with repair processes after mechanical or chemical challenge, or ionizing radio damage[31,41,42]. Following mechanical rupture, the nuclear lamina becomes restored by an endosomal sorting complexes required for transport-dependent manner within 30–90 min[42]. DNA double-strand breaks trigger DNA damage response (DDR) pathways, which mediate DNA repair by non-homologous end joining and homologous recombination, within hours, followed by cell survival[43]. The data further suggest that the strength of lethal hit delivery per CTL contact varies, as indicated by variability in sublethal hit frequency and lethal vs. nonlethal outcomes on a single CTL or target cell level. Such variability may be caused by differences of perforin expression in CTL, the varying strength of TCR signaling, CTL polarization, and exocytosis efficiency, as well as uptake variability of granule content by target cells[44]. Also, damage repair in the target cell may vary, resulting in lethal outcome or survival despite similar frequency and amount of received perforin events. Repair of sublethal CTL hits may further strengthen or weaken target cell susceptibility to subsequent hits, or induce mutations, not unlike other damage and repair processes[45,46].

The concept of "additive cytotoxicity" predicts that sequential, sub-threshold damage events accumulate and induce target cell death. The "cytotoxic dose," expressed as the amount of damage per time, may result from both the magnitude of each damage event and the frequency of sequential hits (Supplementary Fig. 7). Stabilizing contacts may thus promote CTL efficacy by increasing their chance to deliver multiple sublethal hits during the same interaction and become already effective in tumor subregions of low CTL density. However, due to the transient nature of most CTL–tumor cell interactions in vitro and in vivo[10,12,47,48], the median individual contact time per CTL of 15–30 min (Fig. 5c and Supplementary Figs. 1c–e and 2b–d, g–i) may be insufficient for death induction by most single interactions and, hence more likely to be achieved by serially engaging CTL. Accordingly, in solid tumors, CTL killing efficacy is strongly dependent of local CTL density[18] and on CTL migration capacity[49], which jointly mediate frequent sublethal rather than highly effective 1 : 1 pairings. Intracellular damage accumulation may further be mediated by supramolecular attack particles, delivered by transient CTL and NK cell contacts, releasing autonomous cytotoxic complexes on target cell surfaces[50,51]. Thus, CTL interactions induce variably damaging events, which may become integrated over time in the target cell until apoptosis is induced or recovery achieved (Supplementary Fig. 7).

Besides accumulation of sublethal damage by CTL with identical TCR identity, additive cytotoxicity may result from CTL with different TCR specificities, each delivering variable degree of damage, raising the possibility of polyclonal cooperation during the effector phase. Likewise, additive cytotoxicity may enable cooperation of cytotoxic leukocytes with complementary target-recognition strategies, such as CTL and NK cells[52,53] or opsonin-guided macrophages[54]. The requirement of high local density of antigen-specific CTL for apoptosis induction may further provide a "filter," which limits cell death by accidental miss-targeted CTL. This could, e.g., explain the occurrence of autoreactive CTL circulating in the peripheral blood in humans at low frequency without causing clinical effects of autoimmunity[55,56]. Additive cytotoxicity may thus be a mechanism to fine-tune peripheral immune effector function and limit unintended tissue damage by autoimmunity.

The here established criteria for structural trauma in the plasma membrane, nuclear envelope, and DNA, as well as the metrics for frequency and timing of sublethal events, will aid a classification of additive modules and mechanisms by which immune stimulation and suppression jointly modulate CTL efficacy in the tumor microenvironment. Conceivably, additive cytotoxicity may contribute to the efficacy of immuno- combined with molecular-targeted therapy.

In the B16F10/OVA melanoma model, high CTL density and CTL migration between target cells was a prerequisite for frequent CTL contacts and perforin events in the melanoma cells. By immunotherapy, intratumor CTL density can be increased by activation of immunostimulatory pathways[57], antagonization of inhibitory molecules expressed by tumor cells[58], anti-angiogenic therapy normalizing dysfunctional vessel endothelium[59], or by activating the tumor endothelium[60]. Local CTL accumulation can further be achieved by enhancing local CTL proliferation and/or retention by contract stabilization[12]. For example, additive cytotoxicity could underlie the reduced tumor growth in response to combined anti-CTLA-4 and radiation therapy, which enhances local CTL density and contact duration in breast carcinoma[61].

The here observed positive correlation between contact duration and the number of delivered sublethal events suggest that the efficacy of sublethal hit delivery may be improved by stabilizing CTL interactions, e.g., by bispecific antibodies that connect CTL and target cell, and facilitate repetitive hits[5,62] or by agonistic anti-CD137 antibody[12]. The concept of additive cytotoxicity may further provide new rationales for CAR T- and TCR T-cell design. Instead of designing cells with single-epitope specificity and high individual killing capacity at the expense of "on-target/off-tumor" side effects, higher efficacy may be achieved by designing multiclonal CTL, which target epitope combination and mediate less damage in individual interactions. Lastly, similar to chemo- and radio-sensitizing approaches, tumor cell susceptibility to sublethal CTL damage may be increased by combining complementary treatment modalities to induce DNA stress. As example, adoptive CTL transfer could be combined with DDR inhibitors, which target members of the DNA-dependent protein kinase or the poly(ADP-ribose) polymerase family[63].

In conclusion, our data suggests that serial conjugation and delivery of sublethal hits define the efficacy of CTL effector function, which can be exploited by targeted immunotherapy to increase both single-contact efficacy and cooperation of immune effector cells.

## Methods

**Cell lines and primary cell culture.** Mouse embryonic fibroblast-derived C57BL/6 cells (MEC-1) and MEC-1 cells expressing B7.1 and the OVA-derived CTL epitope SIINFEKL (MEC-1/OVA) or the adenovirus type 5 E1A-derived CTL epitope SGPSNTPPEI (MEC-1/E1A) coupled to a signal sequence, which routes the peptide to the endoplasmatic reticulum, were used as target cells[22,23]. The cells were maintained in RPMI 1640 medium (GIBCO, 21875-034) supplemented with fetalcalf serum (FCS) (10%; SIGMA, F7524), HEPES (10 mM; GIBCO, 15630-056),

2-mercaptoethanol (500 mM), penicillin and streptomycin (100 U/ml, each; PAA, P11-010), sodium pyruvate (1 mM; GIBCO, 11360-039), and non-essential amino acids (0.1 mM; GIBCO, 11140-035).

Mouse B16F10 melanoma cells expressed the OVA-derived CTL epitope SIINFEKL (B16F10/OVA). B16F10/OVA cells were transduced to express histone-2B/mCherry, as described[12]. Using sequential passaging, a low-pigmented subline was derived and unchanged growth, invasion ability, antigenicity, and apoptosis resistance in vitro and in vivo were validated. B16F10 lines were cultured in RPMI 1640 medium (GIBCO, 21875-034) supplemented with FCS (10%; SIGMA, F7524), sodium pyruvate (1 mM; GIBCO, 13360-039), and penicillin and streptomycin (100 U/ml each; PAA, P11-010). To increase antigenic peptide expression, B16F10/OVA cells were stimulated with murine IFNγ (200 U/ml; PEPROTECH, 315-05) for 48 h before co-culture with CTL.

Male HLA-A2 expressing human MV3 and BLM melanoma cells were maintained in Dulbecco's modified Eagle medium (GIBCO, 10938-025) supplemented with FCS (10%; SIGMA, F7524), glutamine (2 mM; LONZA, BE17-605), sodium pyruvate (1 mM; GIBCO, 13360-039), and penicillin and streptomycin (100 U/ml each; PAA, P11-010). Cells were stimulated with human IFNγ (200 U/ml; SIGMA, I3265) for 48 h prior to co-culture experiments.

The identity of B16F10, MCF-7, MV3, and BLM cells was verified by short tandem repeat DNA profiling (IDEXX BioResearch). No respective mammalian or mouse interspecies contamination was detected. Neativity for mycoplasma contamination was routinely confirmed (MycoAlert, Lonza).

**Mice.** Mouse breeding and crossings were performed in the Central Animal Laboratory of the Radboud University Nijmegen, The Netherlands. Mice were housed under specific pathogen-free conditions and 12 h light/12 h dark cycles.

Four- to 6-week-old C57BL/6J mice were obtained from Charles River Laboratories. Mice expressing transgenic enhanced green fluorescent protein (eGFP) under the human ubiquitin C promoter (Jackson Laboratories, C57BL/6-Tg(UBC-GFP)30Scha/J, stock number: 004353) and mice expressing transgenic dsRed under the chicken β-actin promoter (Jackson Laboratories, STOCK C57BL/6-Tg(CAG-DsRed*MST)1Nagy/J, stock number: 006051) were crossed to transgenic OT1 TCR mice (Jackson Laboratories, C57BL/6-Tg(TcraTcrb)1100Mjb/J, stock number: 003831) to breed double-transgenic eGFP/OT1 and dsRed/OT1 mice.

C57BL/6-Prf1tm1Sdz/J mice were purchased at Jackson Laboratories (stock number: 002407) and crossed to double-transgenic dsRed/OT1 mice. Genotyping was performed by PCR following the genotyping protocol recommended by Jackson Laboratory. Mice (6–10 weeks of age) homozygous for mutant perforin-1 were used for experiments.

**Ex vivo isolation and activation of primary CD8+ OT1 CTL.** OT1 CTL were isolated and activated as described[12]. Briefly, splenocytes were isolated from OT1 or double-transgenic eGFP/OT1 or dsRed/OT1 mice. Erythrocytes were depleted by ammonium chloride lysis buffer (0.83% NH₄Cl, 0.1% KHCO₃, 0.37% Na₂ EDTA). Antigen-specific CTLs were obtained by splenocyte culture in 24-well plates for 3 days, with an initial cell density of $2.5 \times 10^5$/ml in the presence of SIINFEKL peptide (0.5 µg/ml). On day 3, cultures were supplemented with IL-2 (100 U/ml; ABD SEROTEC, PMP-38) and maintained for further 24–48 h. Cells were collected on days 4 or 5 using Ficoll gradient centrifugation (AXIS-SHIELD PoC AS, Oslo, Norway). Purity of activated OT1 CTL was determined by flow cytometry and typically exceeded 96% of Vα2+CD8+ CD62LlowCD44hi cells (Vα2, BD Bioscience 560622, 1 : 100; CD8a, BD Biosciences 551162, 1 : 100; CD62L, BD Biosciences 553150, 1 : 100; CD44, CALTAG Laboratories RM5701-3, 1 : 100).

After 3D cytotoxicity co-culture, CTL and surviving target cells were collected by dissolving the collagen matrix with collagenase I (40 U per 96-well; 30 min; SIGMA C0130) and detaching adherent cells or cell clusters using trypsin/EDTA (5 min). Cell fractions were combined, washed (phosphate-buffered saline), and stained with primary AlexaFluor488-conjugated anti-Lamp-1 rat-IgG (BIOLEGEND, 121608, 1 : 25) or IgG Isotype control (BIOLEGEND, 400525, 1 : 25), and additional secondary AlexaFluor488-conjugated donkey anti-rat polyclonal antibody (LIFE TECHNOLOGIES, A21208, 1 : 200). CTLs were gated for intact morphology, viability (propidium iodide exclusion), and dsRed expression (De Novo FCS Express 4). Cell populations were calculated using the histogram subtraction function (FCS Express).

**Activation and culture of primary human SMCY.A2 CTL.** The CD8+ SMCY CTL line was isolated and cultured as described[27]. Briefly, CD8+ CTL were isolated from peripheral blood mononuclear cell (PBMC) obtained from a renal cell carcinoma patient who received allogeneic stem cell transplantation (SCT) and donor lymphocyte infusions. Peripheral blood samples for investigational use were obtained under informed consent by the patient, which was approved by the Medical Ethics Committee of the Radboud University Medical Centre and conform to the principles of the Declaration of Helsinki. CD8+ T cells were maintained in Iscove's modified Dulbecco's medium (IMDM; INVITROGEN, Carlsbad, CA) supplemented with human serum (10%; HS; Sanquin blood bank, Nijmegen, The Netherlands) and expanded by weekly stimulation with PBMC obtained before

SCT. Expanded CTLs ($0.5 \times 10^6$) were maintainned in IMDM/10% HS containing irradiated (80 Gy) recipient EBV-LCL ($0.5 \times 10^6$) and irradiated (60 Gy) allogeneic PBMC ($0.5 \times 10^6$) from two donors, together with IL-2 (100 IU/ml; Chiron, Emeryville, CA) and PHA-M (1 mg/ml; Boehringer, Alkmaar, The Netherlands). SMCY.A2-specific CTL were isolated by fluorescence-activated cell sorting (FACS), using tetramer staining of HLA-A2-restricted SMCY.A2 epitope FIDSYICQV, resulting in >95% purity. Specificity of SMCY.A2 CTL for the male epitope was verified by the lack of cytotoxicity against female HLA-A2-expressing breast carcinoma (MCF-7) target cells (Fig. 1d).

**Three-dimensional cytotoxicity CTL–target cell co-culture and time-lapse microscopy.** A sub-confluent monolayer of target cells was overlaid with a 3D collagen gel (PureCol, concentration: 1.7 mg/ml) containing pre-activated CTL in a 24-well plate. After collagen polymerization (30 min, 37 °C), the culture was overlaid with undiluted CTL growth medium. Cell dynamics were recorded by bright-field time-lapse microscopy (30 s frame interval) for 24–48 h.

The duration, kinetics, and outcome of CTL–target cell interactions were quantified by manual analysis. To distinguish CTL–target cell conjugation from irrelevant passengers, the following criteria were used: CTLs in contact with target cells (i) altered their migration track towards or along the target cell, (ii) slowed migration speed, and (iii) changed morphology, such as spreading on or polarization towards the target cell surface (Supplementary Fig. 1b). For quantification of the frequency of serial killing, only CTLs that remained in focus and could be followed for >12 h were analyzed. CTL–target cell interactions were classified as nonlethal when the target cell survived for at least 3 h after CTL detachment. The >3 h follow-up period was obtained from statistics of the time period between last CTL-induced perforin event and target cell death, indicating 1 h in MEC-1/OVA and 1–2 h in B16F10/OVA as cutoffs. Manual analysis was validated by at least two independent researchers as well as blinded analysis.

**Detection of sublethal damage in target cells.** MEC-1/OVA and B16F10/OVA target cells were lentivirally transduced to stably express the calcium sensor GCaMP6s[29]. Cells were selected for stable expression using blasticidin (10 µg/ml: Life Technologies, R210-01) and FACS-sorted for GCaMP6s positivity. pCDH-NLS-copGFP-EF1-BlastiS coding for the NLS-GFP[31] reporter was a gift from Jan Lammerding (Addgene plasmid #132772; http://n2t.net/addgene:132772; RRID:Addgene_132772). Histone-2B-mCherry-expressing B16F10/OVA cells were lentivirally transduced to express NLS-GFP, selected using blasticidin (10 µg/ml: Life Technologies, R210-01) and FACS-sorted for double-positive GFP and mCherry expression. Apple-53BP1trunc[64] was a gift from Ralph Weissleder (Addgene plasmid # 69531; http://n2t.net/addgene:69531;RRID: Addgene_69531). B16F10/OVA cells were lentivirally transduced to stably express Apple-53BP1trunc, selected for stable construct integration with puromycin (2 µg/ml: Sigma-Aldrich, P7255) and FACS-sorted for Apple-positive cells. Sorted cells were further subcloned and three clones were pooled to achieve uniform reporter expression levels.

*Time-lapse detection of sublethal damage.* CTL–target cell interactions and sublethal damage during interaction were detected by co-registering the fluorescent reporter and dsRed OT1 CTL at the following frame intervals and duration: 8–12 s for up to 12 h (GCaMP6s); 2 min for up to 30 h (NLS-GFP). For monitoring CTL conjugation with Apple-53BP1trunc-expressing target cells, which spectrally overlaps with dsRed2, transmission contrast of unlabeled wt OT1 CTL and fluorescence of Apple-53BP1trunc were recorded at time intervals of 5–10 min for up to 48 h (Leica SP8 SMD Confocal). Excitation was limited to 0.05 mW for each excitation line (561 nm for Histone-2B-mCherry, Apple-53BP1trunc and dsRed2; 488 nm for GCaMP6s and NLS-GFP). Viability of CTL and target cells during long-term imaging was verified by the following criteria: constant CTL migration speed; morphological CTL integrity over time including lack cell death of multiple cells; and unperturbed proliferation, in comparison to bright-field imaging. CTL-associated damage events were identified by manual or semi-automated image segmentation and intensity analysis using ImageJ/FIJI.

Ca²⁺ signals in OT1 CTL and target cells were monitored by spinning-disk confocal microscopy (BD Pathway) during 3D co-culture of Fura2-labeled OT1 CTL with GCaMP6s-expressing B16F10/OVA cells, using frame rates of 95 s for Fura2 (340/380 nm) and 8 s for GCaMP6s excitation (488 nm). To account for phototoxicity and bleaching, imaging periods were limited to 1 h.

**Intravital multiphoton microscopy.** All experiments with mice were approved by the Ethical Committee on Animal Experiments and were performed in the Central Animal Laboratory of the Radboud University, Nijmegen (RU-DEC 2009-174, 2011-298, 2017-0034), in accordance with the Dutch Animal Experimentation Act and the European FELASA protocol (www.felasa.eu/guidelines.php). Tumor size was continuously monitored by fluorescence microscopy, with a humane endpoint for tumors exceeding 1 cm³.

Histone-2B/mCherry-expressing tumor cells ($1 \times 10^5$) were injected into the deep dermis of C57/B16J mice (Charles River) carrying a dorsal skin-fold chamber, as described[12]. Three days after tumor implantation, in vitro-activated dsRed OT1

CTLs ($0.5–1 \times 10^6$) were injected intravenously. Tumor-bearing mice were repeatedly monitored for up to 15 days. Intravital multiphoton microscopy was performed on anesthetized mice (1–3% isoflurane in oxygen). During imaging, mice were maintained on a temperature-controlled stage (37 °C). Blood vessels were visualized by intravenous injection of AlexaFluor750-labeled 70 kDa dextran (2 mg/mouse; Invitrogen).

Tumor volume was obtained at consecutive time points from epifluorescence overview images and calculated as (tumor width)$^2$ × (tumor length) × $\pi/6$.

Intravital microscopy was performed on a customized near-infrared/infrared multiphoton scanner (TriMScope-II, LaVision BioTec a Miltenyi Company, Bielefeld, Germany), equipped with three tunable Ti:Sa (Coherent Ultra II Titanium:Sapphire) lasers and an optical parametric oscillator. Time-lapse recordings of CTL interactions with tumor cells were acquired by sequential scanning of 3D tissue regions with 910 nm (GFP, Al750) and 1090 nm (mCherry, dsRed2, second harmonic generation (SHG)) at a laser power of 30 mW (910 nm) and 60 mW (1090 nm). The sampling rate was 1 frame/2 min and sampling periods 4–8 h. For in vivo visualization of intracellular $Ca^{2+}$ intensity, 3D volumes of $250 \times 250 \times 100$ μm were acquired by simultaneous excitation at 910 nm (GCaMP6s, Al750; 20 mW) and 1140 nm (mCherry, dsRed2, SHG; 30 mW) with a sampling rate of 1 frame/10–15 s for periods of 1–2 h.

**Image processing and quantification**. Images were processed using Fiji/ImageJ (version 1.51)[65] (http://pacific.mpi-cbg.de/wiki/index.php/Fiji). Images were stitched to mosaics using the Stitch Grid/Collection plugin and field drifts during time-lapse recording were corrected using the StackReg plugin and the Correct 3D Drift plugin. For image quantifications, the raw, unmodified images were used. For display purposes, time-lapse recordings were adjusted for bleaching using histogram normalization (Bleach Correction plugin). Background noise was filtered using the Remove Outliners plugin and images were scaled and adjusted for brightness, contrast, and gamma, to enhance visualization.

Intact vs. apoptotic tumor nuclei and CTLs were quantified for each individual image section from 3D stacks of $350 \times 350$ μm and an imaging depth of up to 300 μm (7 μm distance between sections). To preclude repeated counting of the same cell, every third slice per stack was analyzed. Nuclei and CTLs were segmented by Gaussian Blur filtering, automated thresholding (Li algorithm) and Watershed separation of touching objects. Objects were filtered for size (Nuclei: ≥60 μm$^2$; CTL: ≥50 μm$^2$) using the Analyze Particles command and counted per slice. To determine apoptosis and mitosis rates from in vivo samples, apoptotic and mitotic nuclei were counted manually and calculated as the percentage of total nuclei per slice. Tumor subregions were defined manually, as follows: "invasion" for cells outside of the tumor main mass; "edge" for cells located in the tumor mass within 250 μm from the tumor–stroma interface; and "core" for cells located >250 μm from the tumor margin (Supplementary Fig. 5d).

53BP1trunc-Apple foci were analyzed using custom scripts to segment nuclei based on Hoechst counterstaining, followed by the detection and counting of foci per nucleus in the Apple Channel using the Find Maxima plugin in FIJI/ImageJ.

NLS-GFP leakage events were quantified by manual tracking of the tumor nuclei using the Manual Tracking Plugin. The intensity of nuclear GFP was divided by the H2B-mCherry signal to correct for mild focus drifts, which affect GFP intensity. Normalized GFP values per nucleus were plotted over time and analyzed manually for leakage events, in combination with manual inspection of the time-lapse recording.

**Statistical modeling**. Target cell survival, perforin events and the resulting survival probability curves were analyzed using the "survival" and "rms" packages in GNU R. Redundant $Ca^{2+}$-positive perforin events, which may coincide with a directly preceding lethal hit and can be misleadingly interpreted as prior, additive hits, were removed, based on the following considerations. The "stochastic killing" hypothesis predicts perforin events to be equally likely to induce apoptosis; therefore, the number of perforin events received prior to the last event preceding apoptosis is not expected to reduce target cell survival. Conversely, redundant perforin events, which are dispensable for apoptosis, may be induced in rapid sequence by the same or other CTL after the final lethal event was received by the target cell. Such redundant events may result in an overestimation of required hits. To remove such possibly redundant events, we used the following approach. The intervals between the last lethal event and apoptosis were assumed as exponentially distributed, and the mean waiting time was fitted using the maximum likelihood method to describe the data with highest fidelity. The perforin event with highest likelihood of being the terminal event was determined using the fitted distribution. This event was defined as the terminal event, with subsequent perforin events considered as redundant. Despite removing redundant contacts, the survival probability and lag times to apoptosis remained significantly dependent on prior perforin events ($p = 0.00032$ with removal, $p = 0.00003$ without) (Supplementary Fig. 5b).

To simulate stochastic killing, the intervals between perforin events and the lag time between the last perforin event to apoptosis were randomly permuted. Assuming stochastic killing, these permutations are not expected to impact target cell survival probability. However, the permuted data resulted in significantly enhanced killing efficiency compared to the actual killing efficiency derived from live-cell measurements, and the amount of perforin events preceding the final event lacked impact on survival probability (Fig. 5c).

Cell damage repair times were estimated from the perforin event data in target cells using a Cox proportional hazards model based on the following assumptions: (1) each perforin event induces a unit amount of damage to the cell; (2) existing damage decays exponentially; (3) the instantaneous risk of apoptosis (hazard) is proportional to the current amount of damage. The damage was entered into the model as a time-dependent covariate and its coefficient represents the amount of damage dealt by each hit. The decay rate of the exponential function was then estimated by minimizing the negative log-likelihood of the model.

**Statistical analysis**. Unpaired Student's $t$-tests or Mann–Whitney $U$-tests, as appropriate, were applied using GraphPad Prism 8.

**Reporting summary**. Further information on research design is available in the Nature Research Reporting Summary linked to this article.

## Data availability
Further information and requests for resources and reagents should be directed to and will be fulfilled by the corresponding authors. Source data are provided with this paper.

## Material availability
This study did not generate new unique reagents. Source data are provided with this paper.

## Code availability
The analysis of the perforin-mediated $Ca^{2+}$ time-to-event data was performed using the R platform for statistical computing, with some preprocessing performed in Python. The R and Python code used for the statistical analysis and simulations of the $Ca^{2+}$ data can be found at https://github.com/jtextor/2020-ctl-killing.

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

## Acknowledgements

We thank Stephen P. Schoenberger for providing the MEC-1/OVA cell line. pCDH-NLS-copGFP-EF1-BlastiS was a gift from Jan Lammerding (Addgene plasmid #132772). This work was supported by the Dutch Cancer Foundation (KWF 2008-4031) to C.G.F. and P.F., a personal KWF grant to A.Th.d.B.), NWO-Rubicon (019.162LW.020) to B.W., the FP7 of the European Union (ENCITE HEALTH TH-15-2008-208142), NWO-VICI (918.11.626), the European Research Council (617430-DEEPINSIGHT), and the Cancer Genomics Cancer, The Netherlands to P.F. Time-lapse confocal microscopy was enabled by an NWO investment grant (834.13.003).

## Author contributions

B.W. and P.F. designed the experiments, interpreted the data, and wrote the paper. A.Th.d.B. designed and performed experiments. B.W. and E.W. quantitatively analyzed the data. K.B. isolated, characterized, and cultured the human SMCY.A2 CTL. J.T. and R.J.d.B. performed mathematical analyses. H.D. and C.G.F. contributed to data interpretation. All authors read and corrected the manuscript.

## Competing interests

The authors declare no competing interests.

**Additional information**

