## [Peer Review File · Nature Communications]

REVIEWER COMMENTS

Reviewer #2 (Remarks to the Author):

In this revised manuscript, the authors addressed many of the initial questions raised in earlier reviews. However, a central claim of this manuscript relates to the concept of “additive cytotoxicity”, in which multiple serial CTL contacts cooperate to deliver lethal killing of target cells. To support this claim, in Fig. 5b the authors analyzed the survival probability of target cells in relation to the number of perforin events (Ca⁺⁺ flux) prior to the final perforin event that resulted in apoptosis. The authors concluded that because the target cells that have received 2 or more perforin events prior to the final perforin event have drastically reduced survival probability, multiple serial perforin hits contribute to the killing of target cells in an additive manner. However, upon closer inspection of Fig. S4a, from which the analysis in Fig. 5b is based on, it appears that the “killing blow” is usually delivered by an individual CTL in a temporal pattern of at least 2 successive perforin events. This explains why survival probability drops substantially for target cells that have had ≥ 2 perforin events before. What is hidden in Fig. 5b, however, is that there are also numerous such successive perforin events delivered by single CTLs that did not immediately lead to lethal damage (either the target cells survived until experiment endpoint, or for at least a few hours until another CTL comes in and delivers the lethal hit). Similarly, there are also cases where single perforin event killed the target cells with this event taking place with enough time since the imaging session began to make it unlikely that there was a proximate preceding perforin event. As such, this again points to the question of whether lethal damage to target cells is derived from a fraction of CTLs that have an adequate killing capacity to deliver the “killing blow”. For these reasons, the analysis in Fig. 5b is useful but does not support the claim of “additive cytotoxicity”. On the same issue, In Figure S4B, it appears that many deaths are driven by sequential perforin events from a single CTL. This implies that the contact time for the killing CTL is quite long (although it is difficult to actually tell due to the length of the x axis) and that doesn't seem consistent with the “average contact time of 15-30min” as stated by the authors in the discussion. Some formal quantification would be useful here.

Related point – The authors observe single hit killers at low CTL density (~40%) in Figure 5E. In these cases, is death mediated by sequential perforin events from a single CTL?

Do tumour cells that interact with the same CTL for a long period of time always exhibit a high number of perforin events? In other words, is the number of perforin events a function of CTL-tumor cell contact duration time?

It is perhaps worthwhile to analyze the data from Fig. S4a by comparing how many of such “successive perforin events” (e.g. >2 perforin events within 50 minutes) led to apoptosis or survival within the next few hours. Additionally, the authors can also incorporate the strength of Ca⁺⁺ flux signaling to determine if certain perforin events are more severe than the others. Quantifying “Ca⁺⁺” T cells alone may not be sufficient to rule out heterogenous TCR signaling. Even relatively weak stimuli result in calcium influx. Do the authors observe variation in the extent or timing of calcium influx across activated CTLs? Do CTLs need to reach a time-integrated calcium signaling threshold to release perforin? Signaling readouts downstream of calcium (e.g. ppERK) would be more informative.

Another analysis that would be useful given that the imaging experiments typically track the cell interactions for >12 hours is to follow the CTLs that have successfully delivered lethal damage to one target cell and determine if these particular CTLs are more likely to cause the same lethal damage to other target cells than the other CTLs that failed to do so.

Can the authors clarify Figure 1C. What does the blue shading represent? It seems that the lag phase to apoptosis decreases for CTLs with a serial kill number of 3 or 4 (the median appears to decrease and fraction of cells in the lower half of the distribution seems to increase). This result would argue that lag phase to apoptosis decreases over consecutive killing events. However, the authors claim that “The lag phase to apoptosis was neither compromised nor accelerated over consecutive killing events (Fig. 1c), which resulted in a consistent eradication frequency of 1 kill every 2 hours (Fig. S1f). This excludes gain of cytotoxicity by kinetic priming through repetitive antigenic interactions.”

The authors show that 80-85% of CTLs receive TCR-mediated signalling (at least in terms of calcium influx) but only 40% of these activation events result in perforin events in B16F10/OVA tumour cells.

- o How variable is perforin expression from CTL-to-CTL prior to mixing them with the tumour cells? Can this expression variability account for the heterogenous perforin events in the tumours?
- o Related point – what factors dictate the number of perforin events that a single CTL is capable of? I would expect the amount of perforin per CTL to become limiting at some point. Can the authors comment on this?
- o Tumour cells themselves may exhibit cell-to-cell variation in their plasma membrane composition/surface charge (e.g. variable glycocalyx components), which could lead to rapid inactivation of secreted perforin. This form of variation could explain, at least partly, why some tumour cells exhibit perforin events while others do not. The authors don't need to test this point, but it is worth discussing briefly.

The authors use a Cox regression model to estimate the damage half-life (56.7 mins) for a single perforin event. Does the timing of serial CTL-tumour interactions that eventually result in death (Fig 5A and Fig. S4B) fall within this predicted half-life? Figure 4D provides quantification for the timing of sequential perforin events by a single CTL only, not multiple CTLs.

In the methods section, the authors stated that >96% of CD62L-low CD44-hi OT-I cell population is obtained after in vitro activation. However, in Fig. R3d, it is clear that >50% of the in vitro activated OT-I cells expressed high level of CD62L. This is an important point to clarify as CD62L can mark differential states of effector differentiation and may affect the ability of the CTLs to perform their killing functions.

Minor:

- 1) In Fig. 6b, tumor subregions are divided into tumor core, edge and invasion. The definition of these regions should be clarified either in text or graphically.
- 2) A tumor cell line expressing nuclear NLS-GFP was used to visualize protein leakage into cytoplasm during nuclear membrane disruption (Movie S3). It is interesting to note that upon membrane "repair", the GFP signal is quickly lost in the cytosol. Is this because the nuclear membrane is repaired at a faster rate than the plasma membrane?

Reviewer #3 (Remarks to the Author):

This is an interesting and timely paper addressing mechanisms of T cell mediated killing in solid tumours. It's long been clear that cells within solid tumours, including lymphomas, are difficult to kill by CTL and the authors have done a good job of setting this up. They use in vitro and in vivo system and assay Calcium flux in the target, nuclear damage (presumably by granzyme B) and double strand breaks in DNA as ways to assess reversible damage in sub-lethal encounters with CTL to support a model of multiple hits being required. The authors use perforin KO T cells to support that the major pathway they are studying is perforin/granzyme rather than FasL. These aspects are all well done. There are a few caveats to address.

The authors use a cell line called MEC-1, but this actually pulls down a leukemic cell line from google search and it would probably be helpful to people trying to repeat this to cite the original paper generating these cell lines- PMID: 7534797. The general idea that it's an in vitro transformed cell that is highly immunogenic and not selected for immune escape seems reasonable otherwise. However, a caveat to this specific use of this "embryonic fibroblast" line as a control for a melanoma, rather than use a normal melanocyte is that normal melanocytes may have lysosome like organelles that they could use in their defence even without this feature being selected for escape. So MEC and its derivatives might be a good control for a fibrosarcoma, but not exactly for a melanoma. This caveat should be mentioned in discussion of evasion mechanisms.

In this same discussion, the authors reinterpret work on membrane repair from Keefe et al as part of a protective mechanism for the target, whereas, Keefe et al saw this repair mechanism as a critical step in introduction of granzymes into the cytoplasm- so needed for killing, not protection from killing. Other groups have actually begun to see such tumour cell reactions as being part of tumour immune evasion. For example, greater resistance of targets to CTL has been associated with fusion of lysosomes with the plasma membrane on the target cell side of the immunological synapse (PMID: 26940455).

In addition to FAS and classical perforin/granzyme release from dense core granules, it has recently been proposed that there is an alternative perforin positive structure referred to a supramolecular attack particles that is released into the immune synapse and accumulates in the target (PMID: 32381591). It would be reasonable to include SMAPs in a list of candidates that could contribute to cumulative damage in a somewhat different way than envisioned by the authors.

If Figure s7 the authors discuss potential microenvironmental manipulations of tumour cells to enhance killing efficiency. Ruocco et al (PMID: 22945631) combined anti-CTLA-4 and radiation therapy to both increase infiltration and increase the duration of T cell-4T1 breast carcinoma interactions to reduce tumour growth. In this setting, anti-CTLA-4 treatment increased the number of infiltrating T cells, but they moved rapidly in the tumour, whereas radiation therapy increase NKG2D ligand expression on 4T1 cells, which appeared to stabilize interactions. This study didn't engage in the careful analysis of the steps in killing, but it may provide a setting in which the model put forward by the authors could be investigated.

Reviewer #4 (Remarks to the Author):

In this interesting study, P. Friedl and coll. investigated the mechanisms of solid tumor cell killing by cytotoxic T lymphocytes (CTL). They describe a mechanism of 'additive cytotoxicity', by which a time-dependent integration of sublethal damage events, delivered by multiple CTL, occurs in target cells. According to this model, tumor cell death or survival in response to CTL attack depends upon the frequency and duration of the "lytic encounters" with CTL.

Results are derived from a combination of 3D time lapse in vitro and in vivo live cell imaging approaches. In my estimation, the reported observations are interesting, the technical quality of the performed experiments is high, and the presented movies are gorgeous and convincing.

I will initially comment on the Authors' reply to the Reviewers comments and afterwards I will summarize my criticisms.

Reply to reviewers:

The reviewers' criticisms are congruent, constructive and reasonable. The authors addressed most of the points and performed a substantial amount of new experimental work.

I believe that the new data and clarifications provided by the authors convincingly address all points raised by Reviewer 2. Concerning the points raised by Reviewers 1 and 3, I think that the authors successfully addressed several major concerns.

Having said that, I believe that, in spite of the fact that the authors addressed the majority of points and provided results that are individually convincing, I am not sure that the manuscript established definitive evidence for the existence of perforin hit summation in individual target cells. This problem has been raised by the reviewers and, in the revised manuscript, it has been only partially solved.

I believe that, instead of a clear evidence of perforin hit summation mechanisms, the revised manuscript presents many converging clues that all together support the proposed model. Moreover, the molecular mechanisms implicated in the accumulation of cytotoxic signals up to a certain threshold (beyond which an irreversible death process is triggered) are elusive.

Specific points:

- To more convincingly show perforin hits summation, it would be important to exclude that target cells die through a mechanism of bystander killing in which lytic components released during the attack of one cell could diffuse in the culture and contribute to killing of adjacent target cells. The recent observations that CTL and NK can release "packages" of lytic components (SMAPs), that can serve as autonomous killing entities supports this hypothesis

(<https://science.sciencemag.org/content/368/6493/897.abstractand>

<https://www.pnas.org/content/117/38/23717>).

Moreover, it cannot be excluded that dying cells release toxic molecules that might affect the viability of other cells. The more inflammatory types of cell death in particular (such as pyroptosis, necroptosis, etc) but also autophagy and apoptosis release large quantities of intracellular DAMPs such as ATP that can be toxic to bystander cells.

(<https://www.ncbi.nlm.nih.gov/pmc/articles/PMC3857631/> and

<https://pubmed.ncbi.nlm.nih.gov/1988462/>).

A straightforward approach to address this point would be to set up experiments in which MHC Class I molecule expression is silenced in a target cell line (to avoid the possibility that antigenic peptides released by dying cells could bind MHC of bystander cells). Parental cells (loaded with the antigenic peptide) and their MHCneg counterparts should be loaded with two different fluoresce probes (in order to identify them) and cultured at 1:1:1 ratio with antigen-specific CTL. Under these conditions the MHCneg should be unaffected, while antigen loaded parental cells should undergo 'additive cytotoxicity'.

- In my opinion, 'additive cytotoxicity' can be inferred by a number of convincing observations that support the model and exclude alternative mechanisms, but cannot be directly proven. Moreover, the precise molecular pathways that are engaged during the accumulation of damage and ultimately trigger irreversible cell death remain elusive. I suggest that the authors downplay a bit the discussion of their results while defending the novelty and importance of their findings.

- It seems to me that results presented in Figure 1c and in Fig S1d are in contradiction; could the authors please clarify?

- The article is difficult to read. While the Methods section is extremely clear, the main text should be improved for interdisciplinary readers. The authors should explain more clearly the rationale of the experiments and organize the flow of the results in a manner that, for instance, results presented in Fig 3 are not discussed before results presented in Fig 2 etc.

- It is important to indicate in the figure legends of some figures (e.g. Figure 1d, 5b, etc) the number of cells corresponding to each curve.

- I am not sure whether this was indicated or not, but it would be important to describe how contact duration (conjugate formation/detachment) was identified, and whether the scores (that are by nature subjective) were independently validated by different individuals, etc.

Reviewer #2 (Remarks to the Author):

In this revised manuscript, the authors addressed many of the initial questions raised in earlier reviews. However, a central claim of this manuscript relates to the concept of “additive cytotoxicity”, in which multiple serial CTL contacts cooperate to deliver lethal killing of target cells.

We provide evidence that sequential sublethal hits increase the probability of apoptosis induction. We have not claimed that sublethal hits need to originate from different CTL. Indeed, our kinetic analyses *in vitro* and *in vivo* suggest that whether single or multiple CTLs achieve apoptosis induction depends on CTL density (Fig 5e, 6i). At high CTL density, sequential contacts and perforin hit delivery precede death induction and efficacy is high. At low CTL density, single CTL are associated with killing, and efficacy is low.

We explain this point as follows (p. 11): *“At high CTL density (ET 1:2) target cell death was frequent and preceded by multiple CTL interactions, whereas at low CTL density (ET 1:16) infrequent apoptotic events were near-exclusively preceded by single-contact engagements (Fig. 5e). (...) This indicates that high CTL density (“swarming”) enables efficient apoptosis induction by favoring serial perforin-dependent hits, predominantly by different CTL, whereas the much-reduced killing efficacy at low CTL density largely relies upon single encounters.”*

To support this claim, in Fig. 5b the authors analyzed the survival probability of target cells in relation to the number of perforin events (Ca⁺⁺ flux) prior to the final perforin event that resulted in apoptosis. The authors concluded that because the target cells that have received 2 or more perforin events prior to the final perforin event have drastically reduced survival probability, multiple serial perforin hits contribute to the killing of target cells in an additive manner.

However, upon closer inspection of Fig. S4a, from which the analysis in Fig. 5b is based on, it appears that the “killing blow” is usually delivered by an individual CTL in a temporal pattern of at least 2 successive perforin events. This explains why survival probability drops substantially for target cells that have had ≥ 2 perforin events before.

What is hidden in Fig. 5b, however, is that there are also numerous such successive perforin events delivered by single CTLs that did not immediately lead to lethal damage (either the target cells survived until experiment endpoint, or for at least a few hours until another CTL comes in and delivers the lethal hit). Similarly, there are also cases where single perforin event killed the target cells with this event taking place with enough time since the imaging session began to make it unlikely that there was a proximate preceding perforin event. As such, this again points to the question of whether lethal damage to target cells is derived from a fraction of CTLs that have an adequate killing capacity to deliver the “killing blow”. For these reasons, the analysis in Fig. 5b is useful but does not support the claim of “additive cytotoxicity”. On the same issue, In Figure S4B, it appears that many deaths are driven by sequential perforin events from a single CTL.

We agree that our data shows variability regarding the consequences of perforin events. This heterogeneity in response ranges from multiple perforin events without death induction to single events followed by target cell death. This heterogeneity may cause the impression, that only rare potent CTL contribute to killing. Particularly prior to apoptosis induction, we often observe a high-frequency ‘burst’ of calcium influx events, in which damage caused by individual hits may accumulate. Alternatively, such bursts may represent redundant hits, facilitated by already weakened target cell defense mechanism following earlier hits and without contributing further to apoptosis induction.

To directly address the extent to which apoptosis induction is associated with serial sublethal hits by a single or multiple independent CTL, we reanalyzed the individual CTL contacts underlying perforin events in B16F10/OVA cells. Based on our estimate of the damage half-life of 56 min (Fig. 5d), >90% of the damage should be repaired after 240 min. Therefore, we analyzed contacts occurring within 240 min prior to apoptosis events. This showed that approximately 2/3 of the apoptosis events at an E:T ratio of 1:4 – 1:8 were associated with perforin events originating from a single CTL, whereas 1/3 of the events were associated with contacts from 2 or 3 CTLs (26 apoptosis events).

We would like to submit, however, that the concept of “additive cytotoxicity” describes the accumulation of sublethal damage events, irrespective of whether these were delivered by a single or multiple CTL. In our *in vitro* studies at an E:T ratio of 1:4 – 1:8, additive cytotoxicity is more biased towards single CTL delivering the hit series. Here, we titrated the CTL density to moderate numbers, to avoid CTL clustering on target cells which makes it more difficult to assign perforin events to single CTL contacts for quantification. *In vivo*, the balance is shifted towards CTL swarming of locally increased CTL density (E:T ratios 1:2 – 1:1), with single contact durations lasting median 15 min. Here, multiple CTL are typically in contact before target cell death (>86%). Thus, whether single or independent CTL deliver a series of sublethal hits depends on the model, CTL density and, beyond the scope of this work, likely microenvironmental parameters.

In the previous revision, we addressed this point by statistical analysis and tested the hypothesis that rare individual CTL deliver the lethal hit, while additional conjugations are irrelevant. In our data (**Fig. S4a**), more than 90% of the apoptotic cells die within 1 h after the last perforin event, and more than 80% only have contact to a single CTL in the 1.5 hours preceding their death. Thus, if the ‘rare potent CTL’ hypothesis were true, in most cases the last CTL in contact should be the one responsible for apoptosis induction. We tested this by assigning the perforin events to the associated individual CTL and extended our model to allow for stochastically variable damage delivered by the last and by prior CTLs, using a multivariate Cox model. This revealed that calcium events associated with both, the last CTL (hazard ratio: 3.7, $p < 10^{-7}$) and earlier CTL in contact (hazard ratio: 3.0, $p = 0.001$) are important predictors of apoptosis. This analysis shows no statistical benefit of distinguishing between last and prior CTL, indicating that their integrated, additive effects are relevant for target cell death. We also included with the previous revision a simpler Cox model analysis that considers all CTL as equally relevant. This more parsimonious description of the data results in the lowest BIC difference of 6.6, whereas the heterogeneity model which focuses on the last interacting CTL alone shows a BIC difference of 2.8. Thus, from a statistical perspective, our current data offers no support for the hypothesis that rare potent CTL drive the killing of target cells.

To explain this analysis and outcomes better, we modified the text as follows (p. 10): *“To integrate timing and heterogeneity of damage, we extended our model to allow for variable damage delivered by serially engaging CTL. A multivariate Cox model shows that both, perforin events associated with the last interacting CTL (hazard ratio: 3.7, $p < 10^{-7}$) and earlier CTL in contact (hazard ratio: 3.0, $p = 0.001$) are both important predictors of apoptosis. This analysis shows no statistical benefit of distinguishing between last and prior CTL, indicating that their integrated, additive effects are relevant for target cell death. A simpler Cox model that considers all CTL equally relevant is a more parsimonious description (as indicated by the lowest BIC) of the data than models that focus on the last interacting CTL alone (BIC difference: 6.6). In addition, a model which assumes variable damage between prior and last CTL contacts was inferior in explaining the data (BIC difference: 2.8). Thus, from a statistical perspective, our current data offers no support for the hypothesis that a rare lethal hit delivered by one CTL leads to target cell elimination. In aggregate, these results indicate that sequential sublethal events delivered by CTL in a timely manner underlie efficient target cell killing.”*

We appreciate any further editorial suggestion how to express this statistically very robust difference to improve the clarity of the statement.

This implies that the contact time for the killing CTL is quite long (although it is difficult to actually tell due to the length of the x axis) and that doesn't seem consistent with the “average contact time of 15-30min” as stated by the authors in the discussion. Some formal quantification would be useful here.

We apologize if this important point was unclear.

Fig. S4a and 4b show registered perforin events from the start of the recording until the target cell apoptosis or loss of the target cell due to migration out of the field of view. These events originate from single or multiple contacts with variable duration.

The variability of contact duration of each individual CTL contact and lag phase between first CTL contact and target cell death are displayed in Fig. S1c-e and S2b-d, g-i. Here, the median of individual CTL contact lasts 15-30 min, depending on the model, with a total range from a few minutes to several hours. To avoid confusion, we now indicate the related data set, as follows: “...the median individual contact time per CTL ranging from 15 – 30 min (Fig. 3c, S1c-e, S2b-d, g-i), (...)”.

Related point – The authors observe single hit killers at low CTL density (~40%) in Figure 5E. In these cases, is death mediated by sequential perforin events from a single CTL?

We agree that this is an interesting question. Monitoring perforin-event frequency in cultures with low CTL density would clarify if single CTL kills were based on repetitive perforin hits in prolonged interactions or on single particularly potent perforin-events. At low CTL densities, killing might further be restricted to particularly sensitive target cell subsets, which would also appear as one-hit events in the analysis.

We would like to submit, that either outcome, single- or multi-hit killing, is in line with the proposed additive cytotoxicity concept: because single hits strong enough to kill a target cell are considered, experimentally and conceptually. However, single, or even dual hits were statistically less likely to induce death compared with 3 and more hits (Fig. 5b). These two scenarios describe both extremes of additive cytotoxicity but are not in disagreement with the model.

Performing the requested experiments of perforin-pore monitoring using the Ca²⁺ reporter at low CTL density to capture single CTL kills is exceedingly difficult because at low CTL density apoptosis is a rare event (ca. 1 apoptosis / 60 target cells within 16 h time-lapse at ET ratio of 1:16). Ca²⁺ imaging further requires high frame rates (10-12 s) and sensitive detection at sufficiently high resolution (40x) which limits the number of fields which can be scanned in sequence, and, thus, number of target cells which can be captured in one experiment.

Target cells per position:	30
Positions per experiment:	4
Apoptotic events per experiment:	2
Time-lapse duration:	16 h

Thus, a minimum of 30 apoptotic events for statistical analysis would require 240 h of time-lapse confocal microscopy (assuming no dropout due to technical challenges such as focus drifts which require a retake of the experiment).

Instead, we reanalyzed the dataset displayed in Fig. S4a and determined the number of perforin-events delivered by a single CTL in single-interaction kills. This showed that 60% of kills were mediated by repetitive hits, while 40% of kills were achieved with one hit (**new Fig. S4c,d**). Thus, also rare single CTL cytotoxic interactions display heterogeneity of calcium events prior to target cell death. We have included this data in the manuscript.

Do tumour cells that interact with the same CTL for a long period of time always exhibit a high number of perforin events? In other words, is the number of perforin events a function of CTL-tumor cell contact duration time?

We reanalyzed the data derived from Ca²⁺ monitoring in cocultures of B16F10/OVA target cells and OT1 CTL and plotted contact duration in correlation to the number of perforin events per contact. We find indeed a correlation between longer contact duration and the number of perforin events delivered by the same CTL (Spearman r 0.3717, $p < 0.0001$) (**Fig. S4e**). The correlation between number of perforin events and contact duration is more prominent in lethal (Spearman r 0.4151, $p < 0.0046$) than in nonlethal contacts (Spearman r 0.3002, $p < 0.0151$). We are prepared to include this data in the manuscript if the reviewers and editor think, it will improve the argument.

To further comply with this request, we analyzed the data from Fig. S4a by comparing “successive perforin events” occurring within the predicted damage half-life time of 56.7 minutes for apoptosis or survival within the next few hours (**Fig. R1**).

Figure R1. Target cell survival probability in dependence of the frequency of calcium events (perforin hits) within 56.7 min. The dots represent censoring times at which cells left the imaging region or reached the end of the imaging timeframe).

Taking the time point of 2 hours after the first calcium event as example, 21%, 30% and 50% of the target cells die within 2 hours after the respective 1, 2 or 3 serial calcium events (**Fig. R1**). Thus, as expected from the data shown in Fig. S4a, cells were less likely to survive after consecutive calcium events rather than a single calcium event.

While this data is didactically intuitive and consistent with the concept that indeed frequent sublethal hits accelerate CTL-mediated killing, we would like to emphasize that this analysis is misleading. A similar pattern as shown in **Fig. R1** can be expected even in the “stochastic killing” model where the damage incurred by the calcium events is not additive: cells that receive several hits in rapid succession would have a higher death rate even if non-lethal damage were cleared instantly. Therefore, this analysis is inferior to the one shown in Fig. S5b and Fig. S4b, which is specifically designed to detect whether additional events prior to death have more impact than expected under the “stochastic killing” model. We therefore feel that this potentially misleading analysis (**Fig. R1**) should not be shown in the paper.

Additionally, the authors can also incorporate the strength of Ca^{++} flux signaling to determine if certain perforin events are more severe than the others.

We agree that quantification of the strength of individual perforin-events would greatly improve our understanding of additive effects. However, we have no indication to believe that the intensity of the GCaMP6s signal in the target cell correlates with intracellular damage strength induced by the perforin event. The dose of granzymes entering through the pores depend on many additional target cell-intrinsic and environmental factors^{1,2}. We therefore believe that GCaMP6-intensity is not a suitable parameter to quantify the strength of individual perforin-events.

Quantifying “ Ca^{++} ” T cells alone may not be sufficient to rule out heterogenous TCR signaling. Even relatively weak stimuli result in calcium influx.

Do the authors observe variation in the extent or timing of calcium influx across activated CTLs?

We agree and expect a certain variability of TCR signaling and intracellular response in both CTL and tumor cells during individual CTL-tumor cell encounters. By quantifying the % of Ca^{2+} positive CTL in target cell contacts, we have ruled out a high prevalence of non-responsive interactions (below 10%) (**Fig. 3a**). We further agree that in natural models with variable TCR subsets significant variability of CTL stimulation and, hence, effector activity can be expected.

To address this concern further, we reanalyzed the timing of Ca^{2+} events in CTL that did respond (**Fig. R2**). All Ca^{2+} events occurred within 5 min after contact initiation (median: 0.6 min), suggesting that TCR triggering occurs reliably and rapidly.

Figure R2. Lag time between CTL-target cell contact initiation and Ca^{2+} event. Median, red line. Data pooled from 3 independent experiments, total of 30 CTL contacts.

We would like to submit, that the proposed concept of additive cytotoxicity is not confounded by the observed variability of contact efficiency at a 1:1 basis. In contrast, it proposes apoptosis induction to occur as a function of (variable) TCR signaling and transmitted damage to the target cell versus recovery phases which are integrated over time.

Do CTLs need to reach a time-integrated calcium signaling threshold to release perforin? Signaling readouts downstream of calcium (e.g. ppERK) would be more informative.

We agree and consider that TCR signaling readout would be highly interesting to better describe the heterogeneity of perforin hit induction upon contacts with target cells. This also would help to understand how to tune CTL-mediated damage therapeutically. However, ppERK staining requires fixation and permeabilization and therefore cannot be combined with the live-cell imaging required to detect perforin events in target cells. Beyond this technical hurdle, we feel that clarifying the signaling events underlying differential perforin release by CTL goes beyond the scope of the current study, which focusses on the identification and accumulation of sublethal damage and its relevance for killing target cells.

Another analysis that would be useful given that the imaging experiments typically track the cell interactions for >12 hours is to follow the CTLs that have successfully delivered lethal damage to one target cell and determine if these particular CTLs are more likely to cause the same lethal damage to other target cells than the other CTLs that failed to do so.

The currently available datasets allow to monitor the same target cell for >12 hours; however, the image sequences do not allow long-term subset analyses of migrating CTL, which frequently move out of the imaging field during monitoring or enter the 3D collagen phase, which both precludes reliable continuous tracking of the same CTL over 12 hours. Subset heterogeneity of CTL and of target cells in coping with sublethal damage would require the acquisition and extensive analysis of even longer movies, larger imaging fields and increased 3D scan depth while co-registering perforin events by Ca^{2+} influx.

To address this relevant point with the available material, we reanalyzed our brightfield long-term imaging data sets, in which we registered serial killing activity of single CTL (data of Fig. 1c). This shows that lag phases before apoptosis induction show a high variability of the lag phases achieved by the same CTL over consecutive lethal interactions (**Fig. R3**). Thus, using the OT1 model which depends on a single transgenic TCR, we obtained no evidence for CTL with consistently very short lag phase and, hence, particularly increased potency.

Figure R3. Variability of lag phase from first CTL contact to MEC-1/OVA target cell apoptosis in serial kills. Lag phases of serial kills are plotted for individual OT1 CTL (one dot resembles one kill). Data of individual CTL are grouped according to the number of observed serial kills (33 CTL from 8 independent experiments).

Can the authors clarify Figure 1C. What does the blue shading represent? It seems that the lag phase to apoptosis decreases for CTLs with a serial kill number of 3 or 4 (the median appears to decrease and fraction of cells in the lower half of the distribution seems to increase). This result would argue that lag phase to apoptosis decreases over consecutive killing events. However, the authors claim that “The lag phase to apoptosis was neither compromised nor accelerated over consecutive killing events (Fig. 1c), which resulted in a consistent eradication frequency of 1 kill every 2 hours (Fig. S1f). This excludes gain of cytotoxicity by kinetic priming through repetitive antigenic interactions.”

We removed the blue shading in **Fig. 1c** (which we agree was unnecessary) and added the p value to compare lag phases to apoptosis between serial kills. The means do not differ significantly (Kuskal Wallis test) and the observed mild decrease of lag phase duration for kill number 3 and 4 are within range of natural variability.

The authors show that 80-85% of CTLs receive TCR-mediated signalling (at least in terms of calcium influx) but only 40% of these activation events result in perforin events in B16F10/OVA tumour cells.

- How variable is perforin expression from CTL-to-CTL prior to mixing them with the tumour cells? Can this expression variability account for the heterogenous perforin events in the tumours?
- Related point – what factors dictate the number of perforin events that a single CTL is capable of? I would expect the amount of perforin per CTL to become limiting at some point. Can the authors comment on this?
- Tumour cells themselves may exhibit cell-to-cell variation in their plasma membrane composition/surface charge (e.g., variable glycocalyx components), which could lead to rapid inactivation of secreted perforin. This form of variation could explain, at least partly, why some tumour cells exhibit perforin events while others do not. The authors don't need to test this point, but it is worth discussing briefly.

We agree that it would be interesting to clarify the mechanisms of heterogeneity of effector function in CTL subsets. However, we would like to submit that addressing these questions experimentally will require extensive in-depth analyses of not only variability of perforin expression in CTL, but also TCR expression, exocytosis kinetics and efficiency in CTL, and the uptake variability of granule content by target cells. We have initiated experiments along these lines, however, anticipate completion will take 1-2 more years. While the origins of heterogeneity in CTL effectivity or variable target susceptibility remain to be clarified, which is a major outstanding theme of the immunology field, we would like to submit that the sublethal hit concept already considers damage of varying intensity. Therefore, requesting these experiments would jeopardize timely publication of this already extensive work.

To discuss the aspect of heterogeneity in CTL signaling and target cell response, we added the following paragraph (p. 14): *“The data further suggest that the strength of lethal hit delivery per CTL contact varies, as indicated by variability in sublethal hit frequency and lethal vs. non-lethal outcomes on a single CTL or target cell level. Such variability may be caused by differences of perforin expression in CTL, the varying strength of TCR signaling, CTL polarization and exocytosis efficiency, as well as uptake variability of granule content by target cells⁴⁴. Also, damage repair in the target cell may vary, resulting in lethal outcome or survival despite similar frequency and amount of received perforin events. Repair of sublethal CTL hits may further strengthen or weaken target cell susceptibility to subsequent hits, or induce mutations, not unlike other damage and repair processes^{45,46}”*

The authors use a Cox regression model to estimate the damage half-life (56.7 mins) for a single perforin event. Does the timing of serial CTL-tumour interactions that eventually result in death (Fig 5A and Fig. S4B) fall within this predicted half-life? Figure 4D provides quantification for the timing of sequential perforin events by a single CTL only, not multiple CTLs.

Using a Cox regression model, we indeed estimate the half-life of a single perforin event in the B16F10/OVA cell line to be 56.7 min. This further predicts, that after ca. 4h >90% of damage is fully recovered by the target cell. Serial CTL encounters of multiple CTL must therefore occur within the 4h prior to apoptosis. As described above, we reanalyzed the data and find that in 1/3 of the apoptosis events, 2 or 3 CTL are in contact within this time period. We feel that the results are sufficiently described in **Fig. 5a**, therefore we have not expanded on this additional plausibility control.

In the methods section, the authors stated that >96% of CD62L-low CD44-hi OT-I cell population is obtained after in vitro activation. However, in Fig. R3d, it is clear that >50% of the in vitro activated OT-I cells expressed high level of CD62L. This is an important point to clarify as CD62L can mark differential states of effector differentiation and may affect the ability of the CTLs to perform their killing functions.

We routinely monitor the CD62L status of the naïve CD8 T cells before and downregulation after SIINFEKL-specific activation and expansion. It is correct, that in the particular experiment shown in the previous rebuttal letter, Fig. R3d, CD62L downregulation was incomplete. However, typically we observe a robust downregulation of CD62L during activation, as shown as example in **Fig. R4**.

Figure R4. CD62L expression on naïve CD8⁺Vβ5⁺ splenocytes (gray) and activated CD8⁺Vβ5⁺ T cells on day 5 of activation (orange).

As presented with the previous revision, we have no indication to support the concern that the used activation protocol results in significant CTL subsets with killing deficiency. The expression of LAMP-1 on the cell surface in 85% of OT1 CTL after co-incubation with B16F10/OVA cells, indicating that most CTL degranulate in response to B16F10/OVA target cells within 30 h of 3D coculture (**Fig. 1b**). As further indication for lack of hypo-responsiveness, 85% of the CTL monitored by time-lapse microscopy kill at least one MEC-1/OVA cell during a 24 h observation period, with at least 50% of CTL acting as serial killers (**Fig. S1f**). As orthogonal reference for physiological relevance of contact dynamics and killing efficacy of the OT1 system towards B16F10/OVA cells, human SMCY-CTL clones confronted with human melanoma cells showed near-identical interaction dynamics and target cell killing (**Fig. S 2e-i**).

Minor:

1) In Fig. 6b, tumor subregions are divided into tumor core, edge and invasion. The definition of these regions should be clarified either in text or graphically.

We added lines delineating the tumor subregions in Fig. S5d and included a better description of the definition of subregions in the methods section (p. 6): *“Tumor subregions were defined manually, with ‘invasion’ including cells outside of the tumor main mass, ‘edge’ including cells in the tumor mass but within 250 μm to the surrounding stroma and ‘core’ including all cells with > 250 μm distance to the tumor-stroma interface (Fig. S5d).”*

2) A tumor cell line expressing nuclear NLS-GFP was used to visualize protein leakage into cytoplasm during nuclear membrane disruption (Movie S3). It is interesting to note that upon membrane “repair”, the GFP signal is quickly lost in the cytosol. Is this because the nuclear membrane is repaired at a faster rate than the plasma membrane?

The perforin pore is repaired rapidly within a few minutes, while relocation of the NLS-GFP signal from the cytoplasm into the nucleus takes on average 50 min (**Fig. 2c**). This is in line with reuptake of NLS-GFP into the nucleus after rupture and repair of nuclear membranes after mechanical trauma⁴. Besides lamins, granzyme B targets e.g., importins, and thus may impact repair mechanisms. Because of the pleiotropic action of granzymes, additional perturbation of NLS-GFP relocation into the nucleus and membrane damage may occur during a CTL contact. Once lamina and associated cytoplasmic-nuclear import machinery are restored, nuclear translocation of NLS-GFP is a rapid process which shows saturation approximately 15 min after microinjection of NLS-GFP into the cytoplasm⁵.

For reasons of space, we have not included this in-depth discussion in the manuscript but will be happy to do so should the referees and editor be supportive.

Reviewer #3 (Remarks to the Author):

This is an interesting and timely paper addressing mechanisms of T cell mediated killing in solid tumours. It's long been clear that cells within solid tumours, including lymphomas, are difficult to kill by CTL and the authors have done a good job of setting this up. They use in vitro and in vivo system and assay Calcium flux in the target, nuclear damage (presumably by granzyme B) and double strand breaks in DNA as ways to assess reversible damage in sub-lethal encounters with CTL to support a model of multiple hits being required. The authors use perforin KO T cells to support that the major pathway they are studying is perforin/granzyme rather than FasL. These aspects are all well done. There are a few caveats to address.

The authors use a cell line called MEC-1, but this actually pulls down a leukemic cell line from google search and it would probably be helpful to people trying to repeat this to cite the original paper generating these cell lines- PMID: 7534797.

We apologize for the confusion and included the original publication of the MEC-1 cell lines⁶, in addition to the citation of the B7.1-expressing variant⁷, which we used in our experiments.

The general idea that it's an in vitro transformed cell that is highly immunogenic and not selected for immune escape seems reasonable otherwise. However, a caveat this this specific use of this "embryonic fibroblast" line as a control for a melanoma, rather than use a normal melanocyte is that normal melanocytes may have lysosome like organelles that they could use in their defence even without this feature being selected for escape. So MEC and its derivatives might be a good control for a fibrosarcoma, but not exactly for a melanoma. This caveat should be mentioned in discussion of evasion mechanisms.

Our main intention was to study sublethal CTL contacts in an independent but 'idealized' model, without aiming to derive knowledge on CTL effector function in a related non-neoplastic model. Using MEC1/OVA cells allowed us to address if sublethal hits and differences in hit frequency occur in an 'easy-to-kill' cell line compared to less sensitive melanoma lines, which indeed can escape immune defense. We have not proposed to conclude that these models represent the natural cause of transformation, and therefore have not included further discussion in the implications for differences in sensitivity during neoplastic transformation. Addressing this question should be reserved to independent work, using a dedicated tumor progression series representing different stage of neoplastic transformation.

In this same discussion, the authors reinterpret work on membrane repair from Keefe et al as part of a protective mechanism for the target, whereas, Keefe et al saw this repair mechanism as a critical step in introduction of granzymes into the cytoplasm- so needed for killing, not protection from killing. Other groups have actually begun to see such tumour cell reactions as being part of tumour immune evasion. For example, greater resistance of targets to CTL has been associated with fusion of lysosomes with the plasma membrane on the target cell side of the immunological synapse (PMID: 26940455).

We thank the reviewer for the insightful discussion. We indeed extended the possible explanations for the data of Keefe et. al. by interpreting the described activation of membrane repair response as possible defense mechanism in other settings. To enhance clarity, we exchanged the reference for Khazen et.al. 2016².

In addition to FAS and classical perforin/granzyme release from dense core granules, it has recently been proposed that there is an alternative perforin positive structure referred to a supramolecular attack particles that is released into the immune synapse and accumulates in the target (PMID: 32381591). It would be reasonable to include SMAPs in a list of candidates that could contribute to cumulative damage in a somewhat different way than envisioned by the authors.

We appreciate this discussion point. The recently discovered SMAPs (perforin/granzyme-containing multiprotein complexes found in stable synapses between CTL/NK cells and target cells) may indeed explain how serial immune cell encounters could deposit cytotoxic molecules on target cell

membranes and induce cytotoxicity upon accumulation. We have included SMAPs in the discussion as follows:

“In addition to intracellular damage accumulation, recently discovered supramolecular attack particles (SMAPs) may contribute to additive cytotoxicity in transient CTL and NK cell contacts by facilitating the accumulation of autonomous cytotoxic complexes on target cell surfaces^{49,50}.”

If Figure s7 the authors discuss potential microenvironmental manipulations of tumour cells to enhance killing efficiency. Ruocco et al (PMID: 22945631) combined anti-CTLA-4 and radiation therapy to both increase infiltration and increase the duration of T cell-4T1 breast carcinoma interactions to reduce tumour growth. In this setting, anti-CTLA-4 treatment increased the number of infiltrating T cells, but they moved rapidly in the tumour, whereas radiation therapy increase NKG2D ligand expression on 4T1 cells, which appeared to stabilize interactions. This study didn't engage in the careful analysis of the steps in killing, but it may provide a setting in which the model put forward by the authors could be investigated.

We thank the reviewer for the valuable input and included the work of Ruocco et al. as potential example for enhanced additive cytotoxicity (p. 16):

“Local CTL accumulation can further be achieved by enhancing local CTL proliferation and/or retention by contract stabilization¹². For example, additive cytotoxicity could underlie the reduced tumor growth in response to combined anti-CTLA-4 and radiation therapy, which enhances local CTL density and contact duration in breast carcinoma⁶⁰.”

Reviewer #4 (Remarks to the Author):

In this interesting study, P. Friedl and coll. investigated the mechanisms of solid tumor cell killing by cytotoxic T lymphocytes (CTL). They describe a mechanism of 'additive cytotoxicity', by which a time-dependent integration of sublethal damage events, delivered by multiple CTL, occurs in target cells. According to this model, tumor cell death or survival in response to CTL attack depends upon the frequency and duration of the "lytic encounters" with CTL.

Results are derived from a combination of 3D time lapse in vitro and in vivo live cell imaging approaches. In my estimation, the reported observations are interesting, the technical quality of the performed experiments is high, and the presented movies are gorgeous and convincing.

I will initially comment on the Authors' reply to the Reviewers comments and afterwards I will summarize my criticisms.

Reply to reviewers:

The reviewers' criticisms are congruent, constructive and reasonable. The authors addressed most of the points and performed a substantial amount of new experimental work.

I believe that the new data and clarifications provided by the authors convincingly address all points raised by Reviewer 2. Concerning the points raised by Reviewers 1 and 3, I think that the authors successfully addressed several major concerns.

Having said that, I believe that, in spite of the fact that the authors addressed the majority of points and provided results that are individually convincing, I am not sure that the manuscript established definitive evidence for the existence of perforin hit summation in individual target cells. This problem has been raised by the reviewers and, in the revised manuscript, it has been only partially solved. I believe that, instead of a clear evidence of perforin hit summation mechanisms, the revised manuscript presents many converging clues that all together support the proposed model.

Moreover, the molecular mechanisms implicated in the accumulation of cytotoxic signals up to a certain threshold (beyond which an irreversible death process is triggered) are elusive.

To accommodate the concern on the level of evidence, we toned down the claim on additive cytotoxicity, stating (on p. 15): "Thus, CTL interactions induce variably damaging events which may become integrated over time in the target cell until apoptosis is induced or recovery achieved." and (on p. 15) "In conclusion, our data suggests that serial conjugation and delivery of sublethal hits define the efficacy of CTL effector function (...)."

Specific points:

- To more convincingly show perforin hits summation, it would be important to exclude that target cells die through a mechanism of bystander killing in which lytic components released during the attack of one cell could diffuse in the culture and contribute to killing of adjacent target cells. The recent observations that CTL and NK can release "packages" of lytic components (SMAPs), that can serve as autonomous killing entities supports this hypothesis

(<https://science.sciencemag.org/content/368/6493/897.abstractand> <https://www.pnas.org/content/117/38/23717>).

Moreover, it cannot be excluded that dying cells release toxic molecules that might affect the viability of other cells. The more inflammatory types of cell death in particular (such as pyroptosis, necroptosis, etc) but also autophagy and apoptosis release large quantities of intracellular DAMPs such as ATP that can be toxic to bystander cells.

(<https://www.ncbi.nlm.nih.gov/pmc/articles/PMC3857631/> and <https://pubmed.ncbi.nlm.nih.gov/1988462/>).

A straightforward approach to address this point would be to set up experiments in which MHC Class I molecule expression is silenced in a target cell line (to avoid the possibility that antigenic peptides released by dying cells could bind MHC of bystander cells). Parental cells (loaded with the antigenic peptide) and their MHCneg counterparts should be loaded with two different fluoresce probes (in order to identify them) and cultured at 1:1:1 ratio with antigen-specific CTL. Under these conditions the MHCneg should be unaffected, while antigen loaded parental cells should undergo ‘additive cytotoxicity’.

The contribution of bystander damage is indeed an important parameter which may enhance additive cytotoxic effects. With the previous revision, we ruled out significant bystander killing mediated by CTL-secreted soluble factors, by adding increasing numbers of perforin-deficient OT1 CTL to cultures with a defined number of wt OT1 CTL (**Fig. S2I**). The reviewer, however, correctly points out that there are other mechanisms which could mediate bystander damage in dying cultures or at least contribute to and partially explain the observed additive effects. These mechanisms include the release of potentially autonomous cytotoxic particles (‘SMAPs’) or toxic molecules originating from dying cell populations and tissue.

To address this possibility, we monitored CTL interactions and killing in mixed cultures of MEC-1/OVA (unlabeled) and the corresponding OVA-negative control cell line (CFSE labeled) (**Fig. R5a**). Tracking OT1 CTL migration in mixed cultures showed that CTL remain focused towards OVA-expressing target cells, resulting in dynamic and long-lasting contacts (90.8 ± 22.8 min) with the CFSE-negative subset, whereas interactions with OVA-negative MEC-1 cells were short-lived (12.8 ± 1.8 min) (**Fig. R5b**). Death events were near-exclusively observed in OVA-expressing target cells (**Fig. R5c**), without obvious indication of bystander damage towards the directly adjacent MEC-1 control cells (see also **Movie R1**, for the discretion of the reviewers). This preliminary data confirms, at single-cell level, that OT1 cells in the 3D collagen model develop low or no bystander damage and confirm the requirement for direct cell-cell contacts for death induction.

To accommodate the potential involvement of SMAPs in additive cytotoxicity in principle, we have included the following statement (on p. 15): “*In addition to intracellular damage accumulation, recently discovered supramolecular attack particles (SMAPs) may contribute to additive cytotoxicity in transient CTL and NK cell contacts by facilitating the accumulation of autonomous cytotoxic complexes on target cell surfaces*^{49,50}.”

Figure R5. Specific cytolytic activity of OT1 CTL towards OVA-expressing target cells. *a*, MEC-1/OVA cells (gray, unlabeled) were seeded with OVA-negative MEC-1/Ctrl cells (green, labeled with 5 μ M CFSE). The mixed cultures were overlaid by a collagen matrix containing OT1 CTL (unlabeled) and monitored by time-lapse microscopy for 12 h (frame rate, 1 min). Zoom in, 0 h: circles, CTL in contact with MEC-1/OVA; 10 h: CTL tracks. *b*, Duration of CTL-target cell contacts. Red bars, median. *c*, Representative images of apoptotic events in MEC-1/OVA target cells. Data from 1 experiment. **Supplementary Movie R1**, Zoom in from R5a. Green, MEC-1/Ctrl cells (5 μ M CFSE); grey, brightfield, MEC-1/OVA and OT1 CTL.

- In my opinion, ‘additive cytotoxicity’ can be inferred by a number of convincing observations that support the model and exclude alternative mechanisms but cannot be directly proven. Moreover, the

precise molecular pathways that are engaged during the accumulation of damage and ultimately trigger irreversible cell death remain elusive. I suggest that the authors downplay a bit the discussion of their results while defending the novelty and importance of their findings.

We have revised the discussion to indicate the converging evidence for additive cytotoxicity by multiple observations and complementary techniques while avoiding an overinterpretation of the results. The changes are indicated in the manuscript in green.

- It seems to me that results presented in Figure 1c and in Fig S1d are in contradiction; could the authors please clarify?

Fig. 1c shows the lag phase until apoptosis derived from contacts of single CTL and plotted over serial encounters with different target cells. Fig. S1d (**revised Fig. S1e**) shows again the lag time until apoptosis, but pools data where the contact duration was derived from single CTL (left dot plot) or multiple, serially interacting CTL (right, cumulative interaction time). The lag phase until apoptosis of single CTL interactions is consistent between Fig. 1c and revised Fig. S1e. To enhance clarity, we updated the figure legend to better indicate that the plot depicts the total interaction time before apoptosis in either single or multi-hit interactions.

- The article is difficult to read. While the Methods section is extremely clear, the main text should be improved for interdisciplinary readers. The authors should explain more clearly the rationale of the experiments and organize the flow of the results in a manner that, for instance, results presented in Fig 3 are not discussed before results presented in Fig 2 etc.

To enhance the readability for interdisciplinary readers, we revised the main text with particular attention to clearly introducing the rationale and aim of each experiment.

In addition, the reviewer noted specifically that the flow of results in Fig. 2 and Fig. 3 should be revised. In Fig. 2 we describe sublethal damage events visualized by 3 distinct reporters, side-by-side. In the text, we first discuss the GCaMP6 reporter for visualizing sublethal perforin-events in target cells (Fig. 2 a-d) and its correlation with Ca^{2+} signaling events in CTL (Fig. 3) before we discuss the other reporters for intracellular damage in greater detail (again Fig. 2 a-d). We acknowledge that this return of the discussion to Fig. 2 makes this passage more difficult to read, however, we consider it important to visualize the quantification of all sublethal damage events and their distinct kinetics side-by-side in the same Figure. We are open to any further editorial suggestion how to resolve this didactic concern and reach maximum readability of the manuscript.

- It is important to indicate in the figure legends of some figures (e.g. Figure 1d, 5b, etc) the number of cells corresponding to each curve.

We apologize for the missing information and have updated all figures accordingly. In addition, we included a detailed Excel file containing the raw data of each graph.

- I am not sure whether this was indicated or not, but it would be important to describe how contact duration (conjugate formation/detachment) was identified, and whether the scores (that are by nature subjective) were independently validated by different individuals, etc.

To improve the clarity of the contact analysis, we added **new Fig. S1B** which visualizes the CTL-target cell contact phases. Contact analysis of CTL-tumor cell interactions in brightfield movies, intravital multiphoton microscopy, as well as Ca^{2+} influx events in CTL and target cells were individually validated by at least 2 researchers for each analysis. We added a statement which describes the validation of the manual analysis in the methods section.

References

1. Nanut, M. P., Sabotič, J., Jewett, A. & Kos, J. Cysteine cathepsins as regulators of the cytotoxicity of nk and t cells. *Frontiers in Immunology* **5**, 616 (2014).
2. Khazen, R. *et al.* Melanoma cell lysosome secretory burst neutralizes the CTL-mediated cytotoxicity at the lytic synapse. *Nat. Commun.* **7**, 10823 (2016).
3. Bird, C. H. *et al.* Cationic Sites on Granzyme B Contribute to Cytotoxicity by Promoting Its Uptake into Target Cells. *Mol. Cell. Biol.* **25**, 7854–7867 (2005).
4. Denais, C. M. *et al.* Nuclear envelope rupture and repair during cancer cell migration. *Science (80-.).* **352**, 353–358 (2016).
5. Rosorius, O. *et al.* Direct observation of nucleocytoplasmic transport by microinjection of GFP-tagged proteins in living cells. *Biotechniques* **27**, 350–5 (1999).
6. Toes, R. E. *et al.* An adenovirus type 5 early region 1B-encoded CTL epitope-mediating tumor eradication by CTL clones is down-modulated by an activated ras oncogene. *J. Immunol.* **154**, 3396–405 (1995).
7. Schoenberger, S. P. *et al.* Efficient direct priming of tumor-specific cytotoxic T lymphocyte in vivo by an engineered APC. *Cancer Res.* **58**, 3094–3100 (1998).

REVIEWERS' COMMENTS

Reviewer #2 (Remarks to the Author):

The authors have responded in detail to the many specific comments and concerns raised in the previous round of review. The new data, new analyses, and re-wording of the paper have substantially improved the submission. It is now clearer to the reader whether the data unequivocally support the concept of additive cytotoxicity or whether there may be several modes of killing, of which this additive behavior is just one. The study represents an extremely careful analysis of the cytotoxic process and the mathematical treatments go far beyond the usual analysis of such data, making this a valuable contribution.

Reviewer #3 (Remarks to the Author):

The authors have addressed my concerns. They have cited the papers describing the cell Mec-1 cell line and accept that they used it as its empirically easy for a CTL to kill. Some of the comments from other reviewers, and particularly reviewer 4 raise issues about the complexity of interpreting the data in a definitive manner vs strongly supporting the model. I feel that the adjustments made to the claims of the manuscript and the issues discussed provide an excellent balanced view of the power and limitations of the results. I congratulate the authors on a heroic dataset and impressive analysis. It's a very important and timely issue in cancer biology that provides a better understanding of that is required to kill highly resistant cells in solid tumours.

Reviewer #4 (Remarks to the Author):

To my opinion, the authors convincingly addressed the comments of reviewers. This is a very interesting and well done study that will have a groundbreaking impact in Immunology.